



# Proportional relationships between carbonaceous aerosols and trace gases in city plumes of Europe and East Asia

Adrien Deroubaix[1,2], Marco Vountas[1], Benjamin Gaubert[3], Maria Dolores Andrés Hernández[1],
Stephan Borrmann[4], Guy Brasseur[2,3], Bruna Holanda[4], Yugo Kanaya[5], Katharina Kaiser[4], Flora Kluge[6],
Ovid Oktavian Krüger[4], Inga Labuhn[7], Michael Lichtenstern[8], Klaus Pfeilsticker[6], Mira Pöhlker[4],
Hans Schlager[8], Johannes Schneider[4], Guillaume Siour[9], Basudev Swain[1], Paolo Tuccella[10], Kameswara
S. Vinjamuri[1], Mihalis Vrekoussis[1], Benjamin Weyland[6], and John P. Burrows[1]

[1]Institute of Environmental Physics, University of Bremen, Bremen, Germany
[2]Max Planck Institute for Meteorology, Hamburg, Germany
[3]Atmospheric Chemistry Observations and Modeling, National Center for Atmospheric Research, Boulder, CO, USA
[4]Max Planck Institute for Chemistry, Mainz, Germany
[5]Research Institute for Global Change, Japan Agency for Marine-Earth Science and Technology, Yokohama, Japan
[6]Institute of Environmental Physics, University of Heidelberg, Heidelberg, Germany
[7]Climate Lab, Institute of Geography, University of Bremen, Bremen, Germany
[8]Deutsches Zentrum für Luft-und Raumfahrt, Oberpfaffenhofen, Germany
[9]Univ Paris Est Creteil, Université Paris Cité, CNRS, LISA, F-94010 Créteil, France
[10]University of L'Aquila, Department of Physical and Chemical Sciences, L'Aquila, Italy

**Correspondence:** Adrien.Deroubaix@iup.physik.uni-bremen.de

**Abstract.**

The concentration of carbonaceous aerosols, black carbon (BC) and organic aerosol (OA), in the atmosphere is related to co-emitted or co-produced trace gases. In this study, we investigate the most relevant proportional relationships between both BC and OA with the following trace gases: carbon monoxide (CO), formaldehyde (HCHO), nitrogen dioxide ($NO_2$), ozone

5    ($O_3$), and sulfur dioxide ($SO_2$). One motivation for selecting these trace gases is that they can be observed using remote sensing measurements from satellite instrumentation, and could therefore be used to predict spatial changes in the amounts of BC and OA.

Airborne measurements are optimal for the analysis of both the composition of aerosols and trace gases in different environments ranging from unpolluted oceanic air masses to those in heavily polluted city plumes. The two aircraft campaigns of the

10    EMeRGe (Effect of Megacities on the Transport and Transformation of Pollutants on the Regional to Global Scales) project have created a unique database, with flight plans dedicated to studying city plumes in two regions, Europe (2017) and East Asia (2018), along with identical instrumental payload.

Using linear regression analysis, three relevant relationships between carbonaceous aerosol and trace gases are identified:

- The BC/OA ratio observed in the Asian campaign is three times higher ($\approx 0.3$) than in the European campaign ($\approx 0.1$),

15    whereas the Pearson correlation coefficient (R) between BC and OA is much higher in Europe (R $\approx$ 0.8) than in Asia (R $\approx$ 0.6).



- The CO/BC ratio is also observed higher in the Asian campaign ($\approx 240$) than in the European campaign ($\approx 170$), whereas the R-value between CO and BC is similar for both campaigns ($R \approx 0.7$).

- The HCHO/OA ratio is similar in both campaigns ($\approx 0.32$), but the observed R-values between HCHO and OA is higher in Europe than in the Asia ($R \approx 0.7$ compared to $\approx 0.3$).

By focusing on heavily polluted air masses sampled downwind in the city plumes, the ratios between the observed carbonaceous aerosols and the five trace gases change, and the R-values increase with $O_3$ for both BC and OA ($R \approx 0.5$).

To assess the performance of atmospheric models with respect to the most relevant observed relationships, an air quality model ensemble is used to represent the current state of atmospheric modeling, consisting of two global and two regional simulations. The evaluation shows that these proportional relationships are not satisfactorily reproduced by the model ensemble. The relationships between BC and OA or between CO and BC are modeled with stronger correlations than the observed ones, and their higher ratios observed in Asia compared to Europe are not reproduced. Furthermore, the modeled HCHO/OA ratio is underestimated in the Asian campaign and overestimated in the European campaign.

This analysis of the proportional relationships between carbonaceous aerosols and trace gases implies that the observed relationships can be used to constrain models and improve anthropogenic emission inventories. In addition, it implies that information about the lower tropospheric concentration of carbonaceous aerosols can potentially be inferred from satellite retrievals of trace gases, particularly in the plumes from megacities.

## 1 Introduction

Although aerosols are ubiquitous components of the atmosphere, their amounts and chemical composition are still largely unknown, not only because of the difficulty of measuring them, but also because of the high spatial variability of biogenic and anthropogenic sources (Prospero et al., 1983; Akimoto, 2003; Li et al., 2022). Accurate knowledge of the amount, composition, and spatial distribution of aerosols is essential to adequately model the Earth system radiative transfer and cloud processes (Kaufman et al., 2002; Li et al., 2022). In addition, this knowledge is required to assess the health impacts of aerosols. This is because some aerosols, such as carbonaceous aerosols, have higher toxicity than others (Lakey et al., 2016).

Carbonaceous aerosols, which we separate in this study into black carbon (BC) and organic aerosol (OA), represent an important part of the aerosol load on a global scale (Szopa and Naik, 2023), and in particular in megacities (e.g., Cheng et al., 2016). BC is a primary aerosol associated with incomplete combustion (Bond et al., 2013), whereas for OA, incomplete combustion is one primary source among others (solvent, industries, waste). The Secondary Organic Aerosols (SOA) formed in the atmosphere by the oxidation of volatile organic compounds (VOC) also contribute to OA (Heald et al., 2008; Jimenez et al., 2009; Zhang et al., 2015; Heald and Kroll, 2020; Li et al., 2022).

Sources of carbonaceous aerosol emissions can be divided into three types: (i) biogenic sources, (ii) forest and agricultural fires, and (iii) anthropogenic sources, for which the emission in cities is high and diverse (Szopa and Naik, 2023), resulting in different ratios of OA and BC and trace gases from one city to another (e.g., Cheng et al., 2016). Megacities, or to a lesser extent all of the "major population centers", which we will simply refer to as cities, represent a special environment because



the emissions of carbonaceous aerosols and trace gases are important and localized and continuous. Even if the amounts of BC and OA are different from one megacity to another (Jimenez et al., 2009; Cheng et al., 2016), we can expect the relative amounts of trace gases co-emitted or co-produced in the atmosphere to be linked for each megacity .

In order to study the relationships between carbonaceous aerosol and trace gases in different environments, measurements are needed. However, there are very few stations measuring continuously both aerosol composition and trace gases such as
the SIRTA (*Site Instrumental de Recherche par Télédétection Atmosphérique*) in France (Zhang et al., 2019). The advantage of aircraft is that they sample different environments (in space), but one disadvantage is that this information is provided as a snapshot (in time). Moreover, the stations provide concentrations at the surface level and the aircraft at various vertical altitudes. Consequently, the connection between surface and vertical-resolved variability needs to be studied. Therefore, aircraft measurement provide a unique opportunity to study these relationships.

The relationships between carbonaceous aerosols and trace gases are probably different over a city and in its plume compared to the surrounding environments, notably due to the oxidation of VOC producing a significant amount of SOA depending on the photochemical activity (Zhang et al., 2015). Several aircraft campaigns have studied the properties of aerosols and have shown that the processes leading to the atmospheric compositions of organic aerosols are only partially understood (Beekmann et al., 2015; Kim et al., 2015). Moreover, analysis of the observational datasets from previous aircraft campaigns in which air
masses were sampled in different environments from clear oceanic air masses to heavily polluted ones, have shown the limited ability of air quality models to reproduce the quantity and variability of carbonaceous aerosols (Wang et al., 2020; Thera et al., 2022).

The two EMeRGe aircraft campaigns (Effect of Megacities on the Transport and Transformation of Pollutants on the Regional to Global Scales), which took place in Europe in 2017 and in East Asia in 2018, are particularly interesting for the study
of the relationships between carbonaceous aerosols and trace gases. This is because the German research aircraft, called HALO (High Altitude and LOng Range Research Aircraft), has carried an identical instrumental payload for both campaigns, with flight plans suitable for studying city plumes (Andrés Hernández et al., 2022; Förster et al., 2023; Lin et al., 2023). In spite of the background concentrations being different during the European campaign (Andrés Hernández et al., 2022) and the Asian campaign (Lin et al., 2023), the aerosol and trace gas measurements made offer a unique opportunity to analyze the statistical
links between the compositions of carbonaceous aerosols and of trace gases, and more specifically having a focus on the city plumes (Förster et al., 2023).

Significant improvements to our knowledge of aerosols and tropospheric pollution have been made by combining aerosol information from multiple satellites (Penning de Vries et al., 2015). For example, over the Amazon, the combustion efficiency has been quantified using satellite products of CO, $NO_2$ and Aerosol Optical Depth (AOD) in order to create a "smoke index"
that could be used to constrain fire emission models (Tang et al., 2019). Over North America, organic aerosols have been estimated by satellite using correlation with HCHO (Liao et al., 2019).

The success of these evolving methodologies, which combine multiple satellite products, implies that they are potentially of great value in the investigation of aerosol composition over megacities. The processes that lead to the production and loss of aerosol are complex at small scales, and can be highly nonlinear. Moreover, observations at these small scales are very



limited. Therefore, we choose to focus on large scales, for which small-scale non-linearities can be smoothed statistically, and for which additional satellite data are available. At these larger scales, the relative amount of aerosols and trace gases can therefore lie within a concise range of variability (for the statistically relevant proportional relationships). Here, we investigate whether there are linear relationships between aerosols and trace gases, and to what extent models are able to reproduce these relationships. This approach is new and sheds more light on the aerosol composition of megacties because the gas and aerosol

emissions are spatially localized and the day-to-day variability is low.

Our hypothesis is that the proportional relationships between carbonaceous aerosols and trace gases are different in megacities and their plumes compared to other environments, consequently we want to understand:

- *What are the relevant proportional relationships between carbonaceous aerosols and trace gases?*

- *Focusing on cities plumes, how are these relationships modified?*

- *Can these relationships be used to improve the satellite-based estimation of aerosols?*

In order to analyze the links between carbonaceous aerosols (*i.e.* OA and BC) and trace gases that are most readily observable by satellites, which are CO, HCHO, $NO_2$, $SO_2$ and $O_3$, we investigate their proportional relationships through a statistical analysis of the measurements of the two EMeRGe aircraft campaigns, and compare them with the relationships reproduced by an air quality model ensemble. The ensemble of air quality models and the identification of city plumes used for the

investigation of two EMeRGe aircraft campaigns are presented in Section 2. The proportional relationships between BC and OA are firstly analyzed (Section 3), secondly between BC and the five trace gases (Section 4), and thirdly between OA and the five trace gases (Section 5). The analysis is extended to the regional scale by studying emission inventories (Section 6). Finally, the relevance of proportional relationships between carbonaceous aerosols and trace gases is discussed by answering the three questions (Section 7).

## 2 The two EMeRGe aircraft campaigns and the air quality modeling

In this section, we present the four simulations that form the air quality model ensemble along with the measurements used from the EMeRGe campaigns (Section 2.1), and the identification of the flight legs corresponding to city plumes (Section 2.2).

### 2.1 Air quality model ensemble

There are different approaches to the use of an air quality model ensemble, depending on the type of analyses planned. In

this study, an ensemble of models is used to represent the state-of-the-art in atmospheric composition modeling. We aim to investigate the proportional relationships between observed concentrations of carbonaceous aerosols and trace gases, and to compare them with the modeled variability of the ensemble.

The air quality model ensemble consists of four simulations, two regional simulations and two global forecasts. The regional simulations are performed with the WRFchem model (Grell et al., 2005; Fast et al., 2006; Powers et al., 2017) at high spatial

resolution (*i.e.* 10 km) over the two regions using the same parameterization (Table A1). To analyze the influence of the meteorological input datasets, two simulations are run for each region with two different meteorological input datasets, one



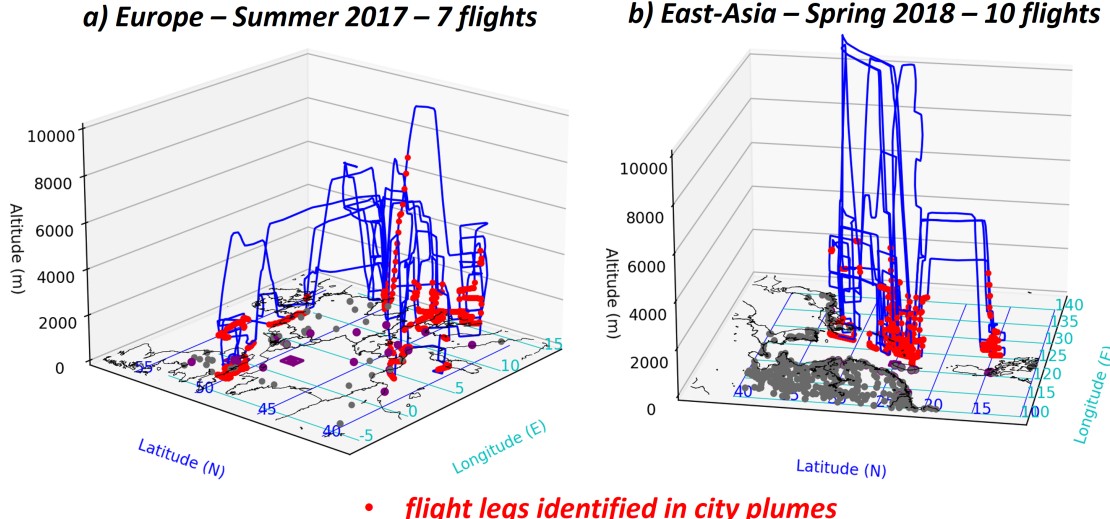

**Figure 1.** *Map of the flights carried out during the two EMeRGe campaigns (a) in Europe and (b) in East Asia. The red dots represent the flight legs identified in city plumes. Cities with more than 500,000 inhabitants are displayed (gray dots). The cities of interest in this study are displayed with purple dots.*

called (i) WRFchem–FNL with the final operational global analysis (FNL) produced by the Global Data Assimilation System of the US National Center for Environmental Prediction (NCEP–FNL) (NCEP, 2022), and the other (ii) WRFchem–ERA5 with the ERA5 reanalysis (Hersbach et al., 2020) produced by the European Centre for Medium-Range Weather Forecasts
(ECMWF–ERA5).

The WRFchem simulations use CAMchem as initial and boundary conditions for the atmospheric composition. This global simulation is provided by the US National Center for Atmospheric Research using the Community Atmosphere Model with Chemistry, which is a component of the Community Earth System Model version 2.1 (Buchholz et al., 2019). To investigate the improvement obtained with the regional simulations, this simulation is also analyzed, called (iii) CAMchem–CESM2. The
global forecast produced by ECMWF through the Copernicus Atmosphere Monitoring Service, hereafter called (iv) CAMS–forecast, is added to the analysis because this forecast at 40 km spatial resolution uses a different configuration for the chemistry and aerosol schemes than the three other models (Table A1). Moreover, CAMS–forecast is produced with the assimilation of several observational datasets (Huijnen et al., 2019; Garrigues et al., 2022; Wagner et al., 2021).

The air quality model ensemble thus consists of four simulations: (i) WRFchem–FNL, (ii) WRFchem–ERA5, (iii) CAMchem–
CESM2, and (iv) CAMS–forecast.



## 2.2 Identification of city plumes during the EMeRGe campaigns

The two EMeRGe campaigns were dedicated to the investigation of the transport and chemical processes that occur in the city plumes. The instrumental payload of the HALO research aircraft enables the same analytical techniques to be applied to the observations made in the two regions, and to assess the similarities and differences of the air masses. The two aircraft

campaigns took place in Europe based in Oberpfaffenhofen, close to Munich (Germany) during the period 11 – 28 July 2017 and in East Asia based in Tainan, close to Taipei (Taiwan) during the period 8 March – 9 April 2018, for which we analyze 7 flights in Europe and 10 flights in Asia. To study the proportional relationships of carbonaceous aerosols and trace gases, the following measurements of meteorology, trace gases and aerosol concentrations are selected from the large set of instruments of the EMeRGe campaigns (Andrés Hernández et al., 2022): (1) Wind speed, (2) CO and (3) $O_3$ (both obtained by UV-photo-

fluorimetry), (4) $NO_2$ and (5) HCHO (both obtained by differential optical absorption spectrometry), (6) $SO_2$ (obtained by chemical ionization mass spectrometry), (7) BC (obtained by photometry), (8) OA (obtained by aerosol mass spectrometry).

For the European and the Asian campaign, the number of 1-min averaged observations studied is 2603 and 4383 respectively, which corresponds to more than 116 hours of sampling. The four simulations of the air quality model ensemble are interpolated in time and in space according to the trajectory of the HALO aircraft in order to generate modeled concentrations

at the locations and time steps of the observations made in the HALO aircraft. The modeled OA corresponds to the sum of primary and secondary organic aerosol. Aerosol concentrations are measured with a cut-off diameter of 1 $\mu$m and the modeled concentrations are compared accordingly.

The identification of city plumes is done using the WRF-CHIMERE model (Menut et al., 2021) by releasing tracers (*i.e.* additional numerical gaseous non-reactive species) emitted at the location of the city centers that are transported by the disper-

sion model. This methodology has been used to study the transport of pollution from major population centers of the Guinean coast (Flamant et al., 2018; Deroubaix et al., 2019; de Coëtlogon et al., 2023) that were investigated during the DACCIWA campaign (Knippertz et al., 2017). To study the datasets of the EMeRGe campaigns, we select the main cities from the two regions, for Europe (Table A2) and for East Asia (Table A3). Flight legs in a city plume are identified using this methodology with the transport induced by NCEP-FNL (NCEP, 2022) and ECMWF-ERA5 (Hersbach et al., 2020), and by retaining flight

legs that are consistent for both meteorological datasets (detailed in Deroubaix et al. (submitted)). We note that most of the city plume legs sampled in EMeRGe correspond to an altitude below 4 km (Figure 1). About 32 % of the flight time is identified in city plumes, which represent 2219 min of observation, and which is consistent with the identification made by Förster et al. (2023) using backward trajectories.

## 3 Proportional relationships between black carbon and organic aerosol

The proportional relationships between BC and OA are investigated for Europe and Asia separately (Section 3.1), and then analyzed focusing on the city plumes (Section 3.2).





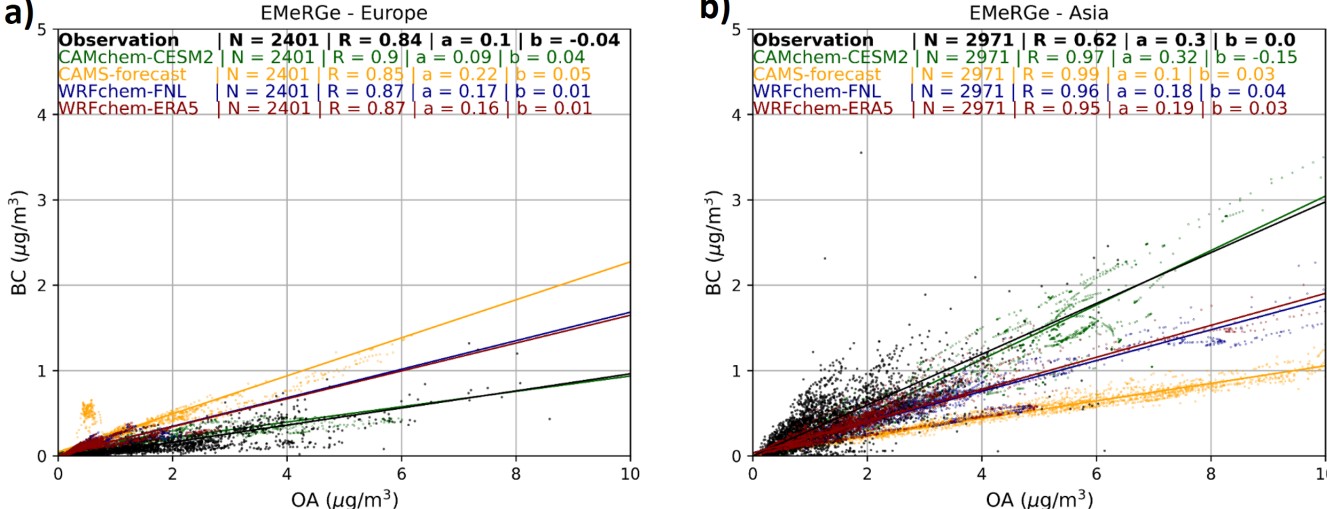

**Figure 2.** *Comparison of black carbon concentrations (BC in $\mu g/m^3$) with organic aerosol (OA in $\mu g/m^3$) during the two EMeRGe campaigns: (a) in Europe and (b) in Asia. Scatter plots of the concentrations of BC against OA for the observations and for an air quality model ensemble composed of two global simulations, CAMchem–CESM2 (green line) and CAMS–forecast (orange line) and two regional simulations, WRFchem–FNL (blue line) and WRFchem–ERA5 (red line). Statistics of linear regression analysis (using reduced major axis regression) are given at the top for each simulation.*

## 3.1 Differences and similarities of the two campaigns

A linear relationship of proportionality exists when there is a strong linear relationship (defined by R), which should be highly statistically significant. If the intercept at the origin is close to zero, then the slope of the linear regression expresses the relative 165 amount between the concentrations of the two species, or more simply their ratios. To compare the proportional relationships of BC and OA for the observations and the air quality model ensemble, linear regressions are performed using reduced major axis regression (the associated slope is denoted *a* and the intercept at the origin is denoted *b*).

According to the observations, the R-value between BC and OA is much higher during the European campaign (R ≈ 0.8) than during the Asian one (R ≈ 0.6), indicating that a proportional relationship is better suited to represent the relative amount 170 between BC and OA in Europe than in Asia at the time of the campaigns (Figure 2). For both campaigns, it is interesting to note that *b* is close to zero for both observations and models. The observed relative amounts of BC compared to OA (*i.e.* the BC/OA ratio) are three times higher in the Asian campaign than in the European campaign, ≈ 0.3 compared to ≈ 0.1 (Figure 2). The R-value of the modeled BC and OA is ≈ 0.9 for the four simulations and for the two campaigns, which is in good agreement with the observed R-value in Europe, but the observed R-value in Asia is lower (R ≈ 0.6). The observed difference 175 in the BC/OA ratio between the two campaigns is well represented by CAMchem–CESM2, and to a lesser extent by CAMS–





forecast, which underestimates the ratio in Asia. The two WRFchem simulations have a similar BC/OA ratio of $\approx 0.17$ for the two campaigns.

In summary, the proportional relationships between BC and OA are observed differently during the European and Asian campaigns, and the linear relationship is better suited during the European campaign than during the Asian campaign. The

high R-values of the air quality model ensemble for both campaigns show that the modeled relationships between BC and OA are much closer to a constant ratio than observed. The three times higher BC/OA ratio observed in Asia can be related to the influence of the wildfires in Indochina (Lin et al., 2023) and could explain the lower performance of the model ensemble in reproducing the variability of the BC/OA ratio.

### 3.2 Focus on the city plumes

For all observations of the two campaigns (Figure A1), the observed linear relationship of BC to OA is associated with a low R-value (R $\approx 0.5$), which is expected because the slope for the European campaign is three times lower than for the Asian one. The air quality model ensemble provide the proportional relationship between BC and OA with stronger R-values than observed (R $\approx 0.9$ compared to R $\approx 0.5$).

Focusing now on the city plumes from both campaigns (Figure A1), the R-value decreases to $\approx 0.3$ for the observed BC/OA

ratio due to the large spread in the scatter plot, while for the air quality model ensemble the linear relationship is still the same (R $\approx 0.9$). Interestingly, the modeled BC/OA ratios are in agreement with those observed (the modeled and observed slopes are $\approx 0.2$), except for the CAMS forecast, which has half the slope of the other three simulations (lower part of the scatter plot).

To conclude, the observed BC/OA ratios are much more variable than the modeled ratios, which are close to a simple linear relationship for all environments sampled by the aircraft. Furthermore, we learn from the observations in city plumes that there

is a larger variability of the BC/OA ratio than for all environments.

## 4 Proportional relationships between black carbon and trace gases

The relative amounts of the five trace gases to BC are investigated by analyzing the most relevant proportional relationships, for Europe and Asia separately (Section 4.1), and then these relationships are analyzed focusing on the city plumes, compared to all observations from both campaigns (Section 4.2).

### 4.1 Differences and similarities of the two campaigns





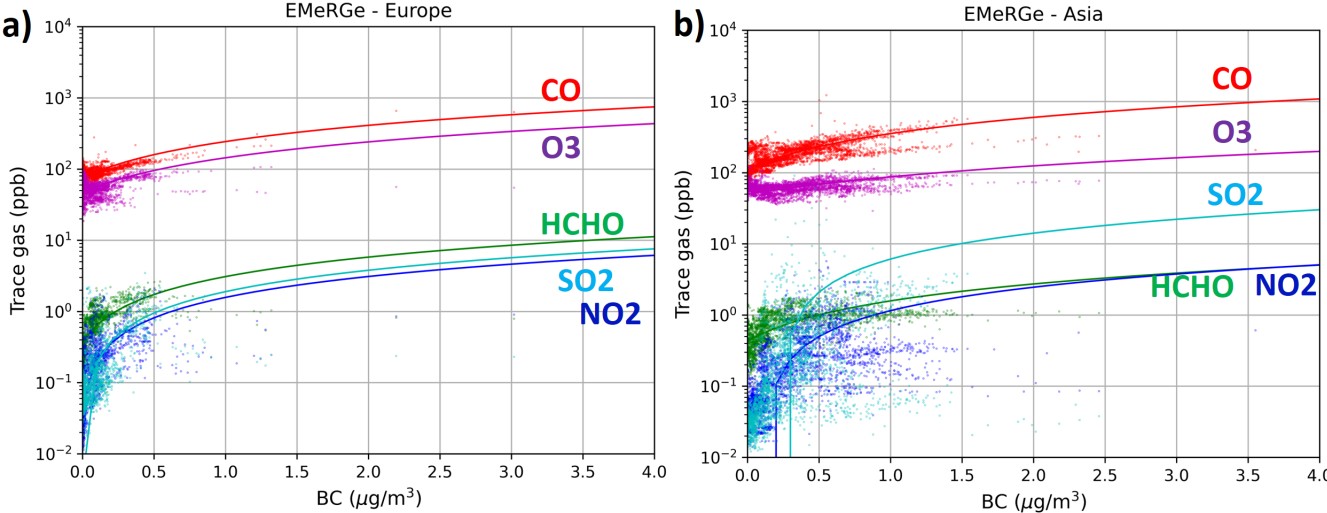

**Figure 3.** *Concentrations of the five trace gases studied (CO, HCHO, NO$_2$, O$_3$ and SO$_2$ in ppb) compared to black carbon (BC in µg/m$^3$) during the two EMeRGe campaigns for the observations (a) in Europe and (b) in Asia: Scatter plots of the concentrations of CO, HCHO, NO$_2$, O$_3$ and SO$_2$ versus the concentration of BC. Colored curves correspond to linear regression analyses (using reduced major axis regression) performed on each trace gas against BC (lines are curved due to the logarithmic Y-axis).*



**Table 1.** *Comparison of linear regression analyses of the five trace gases studied (CO, HCHO, NO$_2$, O$_3$ and SO$_2$ in ppb) against black carbon (BC in µg/m$^3$) during the two EMeRGe campaigns for the observations (left part) in Europe and (right part) in Asia, and for the four simulations of the air quality model ensemble: Correlation coefficients (R), number of points (N), slopes (a) and intercepts (b) corresponding to linear regression analyses (using reduced major axis regression) performed on the concentrations of each CO, HCHO, NO$_2$, O$_3$ and SO$_2$ versus the concentration of BC.*

| Trace gases | R | N | a | b | R | N | a | b |
|---|---|---|---|---|---|---|---|---|
| | **BC - Europe** | | | | **BC - Asia** | | | |
| **EMeRGe observations** | | | | | | | | |
| **CO** | 0.75 | 2569 | 168.01 | 75.36 | 0.65 | 3763 | 243.75 | 110.06 |
| **HCHO** | 0.58 | 1913 | 2.71 | 0.38 | 0.52 | 2442 | 1.15 | 0.43 |
| **NO$_2$** | 0.48 | 1959 | 1.52 | 0.05 | 0.35 | 3056 | 1.31 | -0.15 |
| **O$_3$** | 0.20 | 2563 | 96.81 | 47.19 | 0.42 | 3812 | 37.16 | 50.13 |
| **SO$_2$** | 0.42 | 1949 | 1.89 | 0.00 | 0.15 | 2868 | 8.00 | -1.90 |
| **CAMchem–CESM2** | | | | | | | | |
| **CO** | 0.82 | 2569 | 110.50 | 52.98 | 0.85 | 3763 | 51.88 | 86.68 |
| **HCHO** | 0.75 | 1913 | 6.78 | -0.10 | 0.32 | 2442 | 0.37 | 0.25 |
| **NO$_2$** | 0.43 | 1959 | 1.87 | -0.12 | 0.13 | 3056 | 0.39 | -0.06 |
| **O$_3$** | 0.42 | 2563 | 102.69 | 37.37 | 0.47 | 3812 | 12.25 | 51.76 |
| **SO$_2$** | 0.54 | 1949 | 2.77 | -0.24 | 0.44 | 2868 | 3.46 | -1.42 |
| **CAMS–forecast** | | | | | | | | |
| **CO** | 0.32 | 2569 | 69.48 | 68.95 | 0.63 | 3763 | 182.62 | 113.54 |
| **HCHO** | 0.76 | 1913 | 1.98 | 0.02 | 0.46 | 2442 | 1.54 | -0.16 |
| **NO$_2$** | 0.56 | 1959 | 0.84 | -0.08 | 0.49 | 3056 | 3.51 | -1.05 |
| **O$_3$** | -0.33 | 2563 | -73.22 | 79.10 | 0.23 | 3812 | 31.04 | 53.06 |
| **SO$_2$** | 0.66 | 1949 | 0.54 | -0.07 | 0.70 | 2868 | 3.84 | -0.79 |
| **WRFchem–FNL** | | | | | | | | |
| **CO** | 0.77 | 2569 | 150.20 | 56.20 | 0.92 | 3763 | 129.35 | 86.79 |
| **HCHO** | 0.73 | 1913 | 6.98 | 0.11 | 0.40 | 2442 | 0.76 | 0.19 |
| **NO$_2$** | 0.50 | 1959 | 2.01 | -0.07 | 0.20 | 3056 | 3.34 | -0.94 |
| **O$_3$** | -0.02 | 2563 | -159.33 | 69.91 | -0.02 | 3812 | -20.96 | 71.13 |
| **SO$_2$** | 0.78 | 1949 | 5.14 | -0.17 | 0.37 | 2868 | 18.60 | -4.35 |
| **WRFchem–ERA5** | | | | | | | | |
| **CO** | 0.77 | 2569 | 157.11 | 55.70 | 0.89 | 3763 | 132.16 | 86.90 |
| **HCHO** | 0.75 | 1913 | 7.30 | 0.09 | 0.39 | 2442 | 0.88 | 0.18 |
| **NO$_2$** | 0.53 | 1959 | 1.98 | -0.07 | 0.29 | 3056 | 5.18 | -1.35 |
| **O$_3$** | 0.02 | 2563 | 163.62 | 41.22 | -0.08 | 3812 | -22.18 | 70.40 |
| **SO$_2$** | 0.64 | 1949 | 6.15 | -0.22 | 0.40 | 2868 | 26.62 | -6.16 |



Proportional relationships between observed BC concentrations and observed concentrations of five trace gases are presented using a logarithmic scale (Figure 3) due to the wide range of trace gas concentrations, from CO concentrations greater than 100 ppb to $NO_2$ concentrations less than 0.1 ppb. The linear relationships of BC with the five trace gases are analyzed using reduced major axis regression.

First, the scatter plots of observed concentrations of the five trace gases versus BC shows that the higher BC concentrations in Asia than in Europe are associated with a wider observed concentration range of the five trace gases (Figure 3). The regression lines, which appear as curves due to the logarithmic scale of the Y-axis, are all characterized by positive slopes, which are steeper for CO and for $SO_2$ in Asia than in Europe at the time of the campaigns (Table 1).

The R-values of the observed BC with the observed concentrations of the five trace gases are highest for CO for the two campaigns and of moderate strength ($R \approx 0.7$). The second highest is HCHO with low R-values for the two campaigns ($R \approx 0.5$). The R-values of BC with $NO_2$, $O_3$ and $SO_2$ are less than 0.5 for both campaigns, which means that the observed ratios between these trace gases and BC are very variable. Therefore, there is no relevant linear relationship of $NO_2$, $O_3$ and $SO_2$ with BC during the two EMERGe campaigns.

Looking at the most relevant observed relationships, the analysis of CO/BC and HCHO/BC ratios (*i.e.* the ratios given by the slopes *a*) reveals differences between those in Europe and those in Asia:

- the CO/BC ratio is lower in Europe (110 ppb per $\mu g/m^3$ of BC) than in Asia (168 ppb per $\mu g/m^3$ of BC), therefore 44 % higher during the Asian campaign,

- the HCHO/BC ratio is higher in Europe (2.71 ppb per $\mu g/m^3$ of BC) than in Asia (1.15 ppb per $\mu g/m^3$ of BC), therefore 56 % lower during the Asian campaign.

Second, we compare the R-values and ratios (*i.e.* the slopes *a*) obtained for the air quality model ensemble with those obtained for the observations (Table 1).

In the CAMchem–CESM2 simulation, the CO/BC ratio is higher in Europe (110 ppb per $\mu g/m^3$ of BC) than in Asia (52 ppb per $\mu g/m^3$ of BC), in contrast to the observations, with a relationship closer to linear than observed (R-value of modeled BC and CO is 0.8 compared to 0.7 for observations). The HCHO/BC ratio is significantly higher in Europe (6.78 ppb per $\mu g/m^3$ of BC) than in Asia (0.37 ppb per $\mu g/m^3$ of BC), with a low R-value for the Asian campaign ($R \approx 0.3$ vs. 0.5 for observations).

In the CAMS–forecast simulation, the modeled CO/BC ratio is higher in Europe (69 ppb per $\mu g/m^3$ of BC) than in Asia (83 ppb per $\mu g/m^3$ of BC), with a very low R-value for the European campaign ($R \approx 0.3$ vs. $\approx 0.7$ for observations). The HCHO/BC ratio is modeled in agreement with observations.

In the WRFchem–FNL and WRFchem–ERA5 simulations, the CO/BC ratio in Europe is reproduced (150 ppb and 157 ppb of CO per $\mu g/m^3$ of BC modeled, respectively, compared to 110 ppb per $\mu g/m^3$ of BC observed). In Asia, the R-vale is higher and the slope lower than the observed CO/BC ratio and similar for both campaigns (129 ppb and 157 ppb of CO per $\mu g/m^3$ of BC, respectively, compared to 168 ppb per $\mu g/m^3$ of BC observed). The HCHO/BC ratio is overestimated in Europe (6.78 ppb per $\mu g/m^3$ of BC), whereas it is in agreement with observations in Asia (0.37 ppb per $\mu g/m^3$ of BC). The small differences in




the linear relationships obtained for the WRFchem–FNL and WRFchem-ERA5 simulations suggest a negligible influence of
the meteorological input datasets.

We note the high R-value of BC with $SO_2$ for the model ensemble, which suggest an overestimation of this ratio in the
emission inventory. The CO/BC ratio is modeled with stronger R-values than the observed one, and the higher values of this
ratio observed in Asia compared to Europe are not reproduced. These results show that the proportional relationships of the
five trace gases against BC are not well reproduced by the air quality model ensemble neither during the European nor during
the Asian campaign.

## 4.2 Focus on the city plumes

**Table 2.** *Comparison of linear regression analyses of the five trace gases studied (CO, HCHO, $NO_2$, $O_3$ and $SO_2$ in ppb) against black carbon (BC in $\mu g/m^3$) during the two EMeRGe campaigns for all observations and for the observations that correspond to city plumes: Correlation coefficients (R), number of points (N), slopes (a) and intercepts (b) corresponding to linear regression analyses (using reduced major axis regression) performed on the concentrations of each CO, HCHO, $NO_2$, $O_3$ and $SO_2$ against BC.*

| Trace gases | R | N | a | b |
|---|---|---|---|---|
| **BC - EMeRGe - All observations** | | | | |
| CO | 0.72 | 6332 | 275.88 | 85.35 |
| HCHO | 0.51 | 4355 | 1.46 | 0.41 |
| $NO_2$ | 0.38 | 5015 | 1.24 | -0.05 |
| $O_3$ | 0.37 | 6375 | 46.81 | 49.02 |
| $SO_2$ | 0.18 | 4817 | 6.90 | -1.13 |
| **BC - EMeRGe - City plumes** | | | | |
| CO | 0.67 | 2035 | 306.07 | 64.59 |
| HCHO | 0.31 | 1053 | 1.36 | 0.55 |
| $NO_2$ | 0.22 | 1478 | 1.77 | -0.19 |
| $O_3$ | 0.52 | 2048 | 56.16 | 39.52 |
| $SO_2$ | 0.13 | 1550 | 11.58 | -3.53 |

For all observations of the two campaigns (Table 2), the highest correlation of observed BC is with CO for both campaigns,
which shows a moderate correlation (R ≈ 0.7), followed by HCHO with a low correlation for both campaigns (R ≈ 0.5). The
correlations with $NO_2$, $O_3$ and $SO_2$ have R-values below 0.5 for both campaigns. CO is found to be the best proxy for BC, but
these results are influenced by the consideration of all measurements encompassing various air masses.

Focusing now on the observations in city plumes from both campaigns, the R-value of BC with CO decreases (from ≈ 0.7 to
0.5), as does HCHO (from ≈ 0.5 to 0.2), while the R-value increases with $O_3$ (from ≈ 0.4 to 0.5). These findings indicate that



in city plumes, the most relevant proportional relationships of the five trace gases against BC are different compared to those for all observations.

## 5    Proportional relationships between organic aerosol and trace gases

The relative amounts of the five trace gases to OA are investigated by analyzing the most relevant proportional relationships as in the previous sections, for Europe and Asia separately (Section 5.1), and then these relationships are analyzed focusing on the city plumes, compared to all observations from both campaigns (Section 5.2).

### 5.1    Differences and similarities of the two campaigns

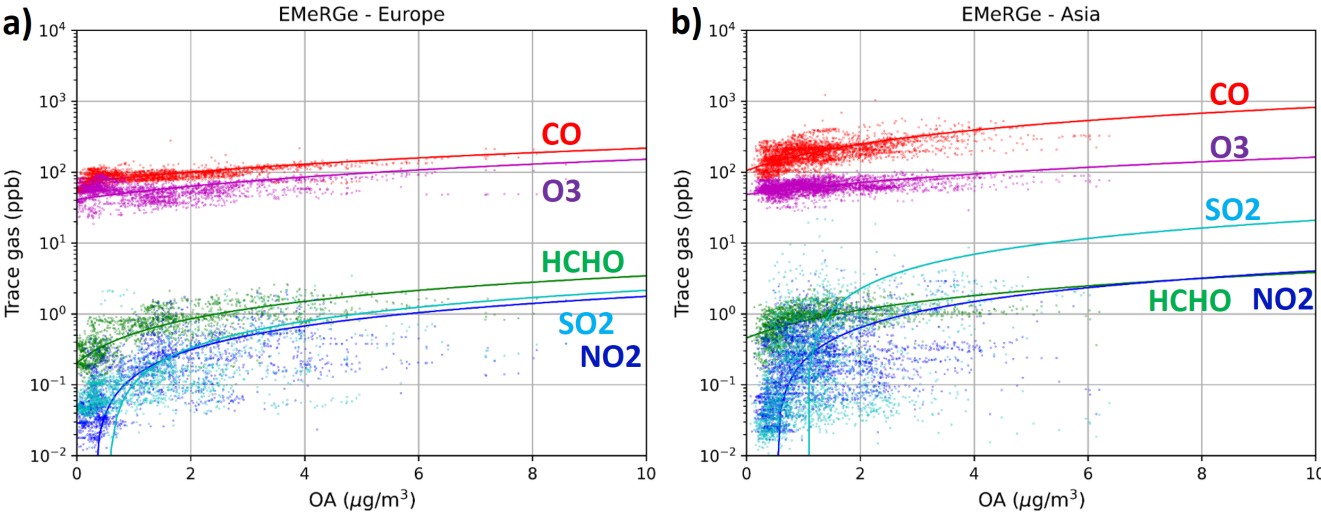

**Figure 4.** *Concentrations of the five trace gases studied (CO, HCHO, $NO_2$, $O_3$ and $SO_2$ in ppb) compared to organic aerosol (OA in $\mu g/m^3$) during the two EMeRGe campaigns for the observations (a) in Europe and (b) in Asia: Scatter plots of the concentrations of CO, HCHO, $NO_2$, $O_3$ and $SO_2$ versus the concentration of OA. Colored curves correspond to linear regression analyses (using reduced major axis regression) performed on each trace gas against OA (lines are curved due to the logarithmic Y-axis).*





**Table 3.** *Comparison of linear regression analyses of the five trace gases studied (CO, HCHO, NO$_2$, O$_3$ and SO$_2$ in ppb) against organic aerosol (OA in µg/m$^3$) during the two EMeRGe campaigns for the observations (left part) in Europe and (right part) in Asia, and for the four simulations of the air quality model ensemble: Correlation coefficients (R), number of points (N), slopes (a) and intercepts (b) corresponding to linear regression analyses (using reduced major axis regression) performed on the concentrations of each CO, HCHO, NO$_2$, O$_3$ and SO$_2$ versus the concentration of OA.*

| Trace gases | R | N | a | b | R | N | a | b |
|---|---|---|---|---|---|---|---|---|
| | **OA - Europe** | | | | **OA - Asia** | | | |
| | **EMeRGe observations** | | | | | | | |
| **CO** | 0.66 | 2373 | 14.67 | 70.65 | 0.50 | 3398 | 71.56 | 105.20 |
| **HCHO** | 0.68 | 1789 | 0.32 | 0.20 | 0.32 | 2106 | 0.34 | 0.46 |
| **NO$_2$** | 0.44 | 1794 | 0.18 | -0.05 | 0.25 | 2856 | 0.43 | -0.21 |
| **O$_3$** | 0.22 | 2367 | 11.04 | 41.15 | 0.33 | 3471 | 11.41 | 48.42 |
| **SO$_2$** | 0.46 | 1789 | 0.23 | -0.12 | 0.11 | 2594 | 2.33 | -2.41 |
| | **CAMchem–CESM2** | | | | | | | |
| **CO** | 0.90 | 2373 | 10.03 | 56.45 | 0.83 | 3398 | 17.05 | 75.87 |
| **HCHO** | 0.82 | 1789 | 0.61 | 0.12 | 0.40 | 2106 | 0.13 | 0.25 |
| **NO$_2$** | 0.35 | 1794 | 0.17 | -0.06 | 0.17 | 2856 | 0.19 | -0.22 |
| **O$_3$** | 0.52 | 2367 | 9.11 | 40.66 | 0.50 | 3471 | 4.60 | 45.98 |
| **SO$_2$** | 0.36 | 1789 | 0.25 | -0.15 | 0.44 | 2594 | 1.01 | -1.61 |
| | **CAMS–forecast** | | | | | | | |
| **CO** | 0.59 | 2373 | 15.53 | 71.59 | 0.64 | 3398 | 18.70 | 118.00 |
| **HCHO** | 0.69 | 1789 | 0.44 | 0.12 | 0.42 | 2106 | 0.16 | -0.09 |
| **NO$_2$** | 0.24 | 1794 | 0.19 | -0.04 | 0.48 | 2856 | 0.42 | -1.16 |
| **O$_3$** | -0.00 | 2367 | -16.16 | 76.23 | 0.32 | 3471 | 3.45 | 51.10 |
| **SO$_2$** | 0.51 | 1789 | 0.12 | -0.04 | 0.72 | 2594 | 0.40 | -0.69 |
| | **WRFchem–FNL** | | | | | | | |
| **CO** | 0.86 | 2373 | 25.35 | 57.66 | 0.89 | 3398 | 23.26 | 91.40 |
| **HCHO** | 0.60 | 1789 | 1.11 | 0.19 | 0.25 | 2106 | 0.17 | 0.22 |
| **NO$_2$** | 0.34 | 1794 | 0.32 | -0.05 | 0.07 | 2856 | 0.63 | -0.98 |
| **O$_3$** | 0.13 | 2367 | 25.98 | 43.20 | 0.02 | 3471 | 4.23 | 52.09 |
| **SO$_2$** | 0.52 | 1789 | 0.80 | -0.10 | 0.24 | 2594 | 3.56 | -4.44 |
| | **WRFchem–ERA5** | | | | | | | |
| **CO** | 0.85 | 2373 | 25.93 | 57.52 | 0.82 | 3398 | 24.64 | 90.85 |
| **HCHO** | 0.62 | 1789 | 1.15 | 0.19 | 0.28 | 2106 | 0.20 | 0.20 |
| **NO$_2$** | 0.34 | 1794 | 0.31 | -0.04 | 0.10 | 2856 | 0.99 | -1.36 |
| **O$_3$** | 0.17 | 2367 | 25.65 | 43.52 | -0.00 | 3471 | -4.61 | 70.42 |
| **SO$_2$** | 0.42 | 1789 | 0.94 | -0.13 | 0.23 | 2594 | 5.13 | -6.05 |





The relationships between the observed OA concentration and the observed concentrations of the five trace gases are presented, as in the previous section on BC, using the scatter plots (Figure 4) and complemented by the analysis of the linear relationships between OA and the five trace gases for the observations and for the air quality model ensemble (Table 3).

First the scatter plots of observed concentrations of the five trace gases versus OA show that a higher concentration of OA is associated with higher concentrations of the five trace gases, and the concentration range is higher in Asia than in Europe

(Figure 4), as noted for BC in the previous section. The regression lines obtained from the observations are also all associated with positive slopes, which are steeper for CO and for $SO_2$ in Asia than in Europe at the time of the campaigns (Table 3).

The R-values of the observed OA with the observed concentrations of the five trace gases are the highest for CO for the two campaigns (R $\approx$ 0.6), however we note that the R-value of OA with HCHO is as high as for CO in the European campaign while it is low in the Asia campaign. The R-values of OA with $NO_2$, $O_3$ and $SO_2$ are less than 0.5 for both campaigns, therefore

there is no linear relationship of these variables with OA during the two EMERGe campaigns, as previously noted in the case of BC. Therefore, there is no relevant linear relationship of $NO_2$, $O_3$ and $SO_2$ with OA during the two EMERGe campaigns, as previously noted in the case of BC.

Looking at the most relevant observed relationships, the analysis of CO/OA and HCHO/OA ratios shows that:

– the CO/OA ratio is lower in Europe (15 ppb per $\mu$g/m$^3$ of OA) than in Asia (72 ppb per $\mu$g/m$^3$ of OA), therefore 4.8

270        times higher during the Asian campaign,

– the HCHO/OA ratio is similar in Europe (0.32 ppb per $\mu$g/m$^3$ of OA) and in Asia (0.34 ppb per $\mu$g/m$^3$ of OA).

Second, we compare the R-values and ratios (*i.e.* the slopes *a*) obtained for the air quality model ensemble with those obtained for the observations (Table 3).

In the CAMchem–CESM2 simulation, the CO/OA ratio from the CAMchem–CESM2 simulation is slightly lower in Europe

($\approx$ 10 ppb per $\mu$g/m$^3$ of OA) than in Asia ($\approx$ 17 ppb per $\mu$g/m$^3$ of OA), with a stronger linear relationship than observed for both campaigns (R-value of modeled OA and CO is 0.9 compared to 0.6 for observations). The HCHO/OA ratio is notably higher in Europe ($\approx$ 0.6 ppb per $\mu$g/m$^3$ of OA) than in Asia ($\approx$ 0.1 ppb per $\mu$g/m$^3$ of OA), while the higher R-values observed in Europe than in Asia are reproduced by the model.

In the CAMS–forecast simulation, the modeled CO/OA ratio agrees with observations in Europe ($\approx$ 16 ppb per $\mu$g/m$^3$

of OA) but underestimates in Asia ($\approx$ 19 ppb per $\mu$g/m$^3$ of OA), with R-values in agreement with the observed ones. The HCHO/OA ratio is similar to the observed one in Europe ($\approx$ 0.4 ppb per $\mu$g/m$^3$ of OA) but is underestimated in Asia ($\approx$ 0.2 ppb per $\mu$g/m$^3$ of OA), while the R-values are in agreement with the observations.

For the WRFchem–FNL and WRFchem–ERA5 simulations, the CO/OA ratios are similar for the two simulations and both campaigns ($\approx$ 25 ppb of CO per $\mu$g/m$^3$ of OA), with R-values greater than 0.8, indicating an overly strong linear relationship

across regional simulations. The HCHO/OA ratio is modeled much higher in Europe ($\approx$ 1.1 ppb per $\mu$g/m$^3$ of OA) than in Asia ($\approx$ 0.2 ppb per $\mu$g/m$^3$ of OA), while the R-values are in agreement with the observations as for the global simulations.



In conclusion, the air quality model ensemble underestimates the HCHO/OA ratio in the Asian campaign and overestimates in the European campaign. These results show that the proportional relationships of the five trace gases against OA are not well reproduced by the air quality model ensemble in the different environments sampled by the HALO aircraft.

## 290    **5.2   Focus on the city plumes**

**Table 4.** *Comparison of linear regression analyses of the five trace gases studied (CO, HCHO, NO$_2$, O$_3$ and SO$_2$ in ppb) against organic aerosol (OA in μg/m$^3$) during the two EMeRGe campaigns for all observations and for the observations that correspond to city plumes: Correlation coefficients (R), number of points (N), slopes (a) and intercepts (b) corresponding to linear regression analyses (using reduced major axis regression) performed on the concentrations of each CO, HCHO, NO$_2$, O$_3$ and SO$_2$ against OA.*

| Trace gases | R | N | a | b |
|---|---|---|---|---|
| **OA - EMeRGe - All observations** | | | | |
| **CO** | 0.29 | 5771 | 67.96 | 62.95 |
| **HCHO** | 0.49 | 3895 | 0.34 | 0.33 |
| **NO$_2$** | 0.27 | 4650 | 0.32 | -0.15 |
| **O$_3$** | 0.25 | 5838 | 11.54 | 45.08 |
| **SO$_2$** | 0.10 | 4383 | 1.64 | -1.77 |
| **OA - EMeRGe - City plumes** | | | | |
| **CO** | 0.10 | 2071 | 70.42 | 54.90 |
| **HCHO** | 0.39 | 1116 | 0.30 | 0.47 |
| **NO$_2$** | 0.09 | 1586 | 0.42 | -0.25 |
| **O$_3$** | 0.46 | 2085 | 12.80 | 38.08 |
| **SO$_2$** | 0.06 | 1584 | 2.52 | -3.86 |

For all observations of the two campaigns (Table 4), the best proxy for OA with one of the five trace gases studied is HCHO with a moderate correlation (R ≈ 0.5) when considering the different environments sampled by the HALO aircraft. It is interesting to note that the CO is not a good proxy for OA for the two campaigns because the ratios are very different (compared to those for BC), which R-value decreases from ≈ 0.6 for each campaign to ≈ 0.3 for both two campaigns.

Focusing now on the observations in city plumes from both campaigns, the R-value of OA with HCHO slightly decreases, while the R-value increases with O$_3$ (from ≈ 0.3 to 0.5). Analogous to the results for BC in the previous section, these results underscore modifications in the most relevant proportional relationships of the five trace gases against OA within city plumes.



# 6 Linking proportional relationships in emission inventories to observed concentrations

The three previous sections have shown that the modeled relative amounts do not agree with those observed for the most rele-
vant proportional relationships (with R greater than 0.5), namely BC/OA (*cf.* Section 3), CO/BC (*cf.* Section 4), and HCHO/OA
(*cf.* Section 5).

Kanaya et al. (2021) have shown large discrepancies of the CO/BC ratio in global anthropogenic emission inventories in
China. The large discrepancies between the modeled and observed concentration ratios suggest that the emission inventories
do not represent these ratios well. Therefore, this section compares ratios derived from observed and modeled concentrations
with those derived from anthropogenic and fire emission inventories.

**Table 5.** *Ratios of BC/OA, CO/BC, HCHO/OA of observed and modeled concentrations by the four simulations of the air quality model ensemble for the two EMeRGe campaigns in Europe and in Asia.*

| *Dataset* | BC/OA | CO/BC | HCHO/OA |
|---|---|---|---|
| **EMeRGe observations** | | | |
| Europe | 0.10 | 168.01 | 0.32 |
| Asia | 0.29 | 243.75 | 0.34 |
| **Model ensemble** | | | |
| *CAMchem–CESM2* | | | |
| Europe | 0.08 | 110.50 | 0.61 |
| Asia | 0.31 | 51.88 | 0.13 |
| *CAMS–forecast* | | | |
| Europe | 0.22 | 69.48 | 0.44 |
| Asia | 0.10 | 182.62 | 0.42 |
| *WRFchem–FNL* | | | |
| Europe | 0.16 | 150.20 | 1.11 |
| Asia | 0.16 | 129.35 | 0.17 |
| *WRFchem–ERA5* | | | |
| Europe | 0.18 | 157.11 | 1.15 |
| Asia | 0.18 | 132.16 | 0.20 |

Several anthropogenic emission inventories are used by the four simulations of the air quality model ensemble, including
CAMS-GLOB-ANTv4.2 (Granier et al., 2019) and EDGARv4.3.2 (Crippa et al., 2018), as well as fire emission inventories,
including GFED4 (Giglio et al., 2013) (*cf.* Table A1). Regional analyses are performed for Europe (July) and Asia (April),
evaluating the spatial variability of the emission flux ratios (Figure 5, Figure A2, and Figure A3).



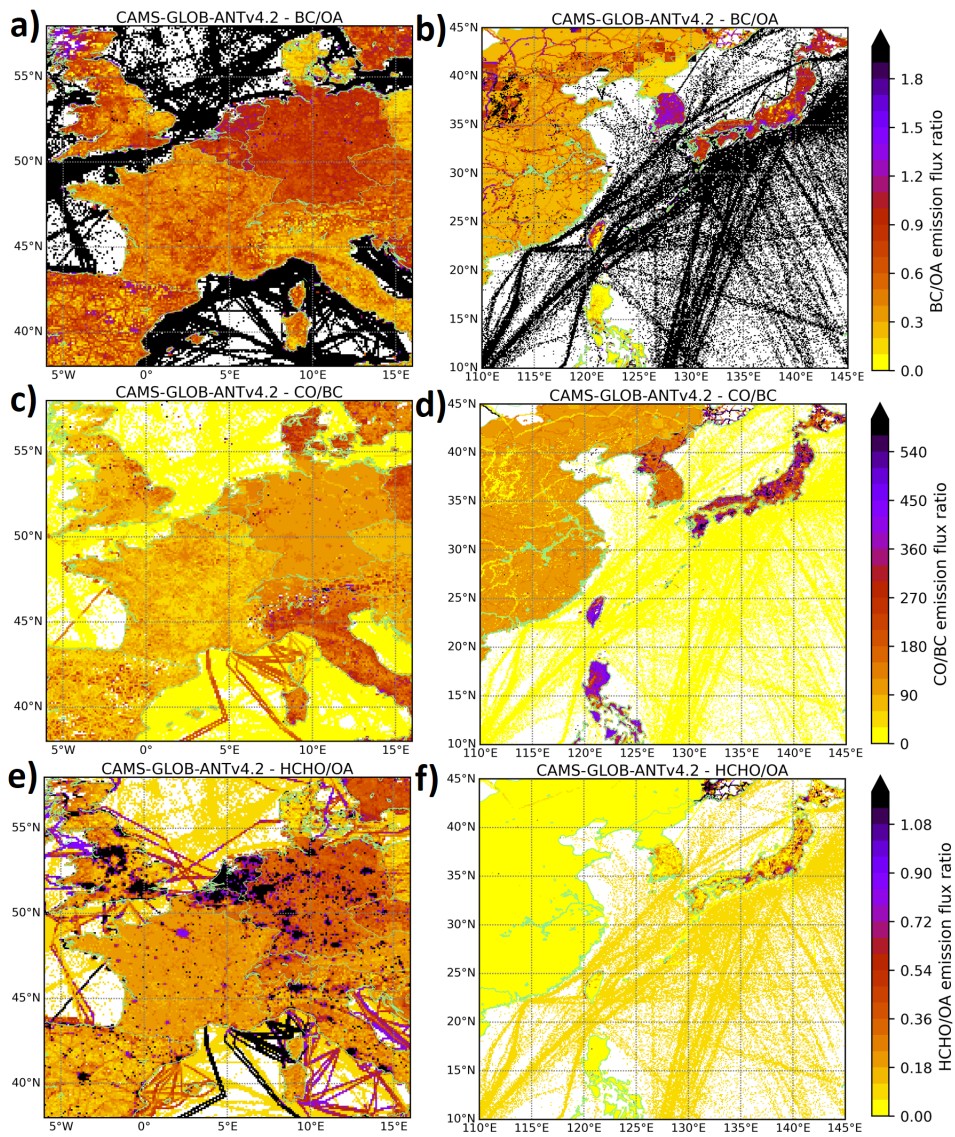

**Figure 5.** *Maps of emission flux ratios from the anthropogenic inventory CAMS-GLOB-ANT v4.2 (sum of all sectors): a) for BC/OA in Europe for July 2017, b) for BC/OA in Asia for April 2018, c) for CO/BC in Europe for July 2017, d) for CO/BC in Asia for April 2018, e) for HCHO/OA in Europe for July 2017, and f) for HCHO/OA in Asia for April 2018.*

The spatial variability of the emission flux ratios is similar for the two anthropogenic inventories, with the BC/OA ratio reaching 2, the CO/BC ratio reaching 600, and the HCHO/OA ratio reaching 1. For both inventories, we note the same large changes in emission flux ratios at the border of two countries, which may be related to the reporting of national emissions. Comparing the fire inventory with the two anthropogenic emission inventories, the BC/OA ratio in GFED4 is higher than the



anthropogenic inventories in Europe and Asia (reaching 10), and the CO/BC and HCHO/OA ratios are lower (reaching 200
and 0.9, respectively).

The emission flux ratios in Europe and Asia are compared with the observed and modeled concentration ratios derived from
the slopes $a$ determined in the previous sections (summarized in Table 5). Emission flux ratios are expected to be different
from atmospheric concentration ratios due to differences in transport and dilution, chemical reactions, and deposition of the
two species considered. Nevertheless, there are notable differences in the observed concentration ratios between Europe and
Asia that are inconsistently reproduced by the model. If the differences in the modeled concentration ratios between the two
campaigns are consistent with the differences in the emission flux ratios, but inconsistent with the observations, then it suggests
that the same inconsistencies exist in the emission flux ratios.

Notable differences in the observed concentration ratios between Europe and Asia include:

1) For the BC/OA ratio, the observed value is 3 times higher in Asia than in Europe. Conversely, for both anthropogenic
inventories, the BC/OA ratio is higher in Europe than in Asia (ranging from 0.5 to 1 in Europe and from 0.2 to 1.5 in Asia),
which can explain the modeled BC/OA ratio underestimation by WRFchem simulations and CAMS–forecast in Asia (Figure
5 and Figure A2).

2) For the CO/BC ratio, the observed value is 1.5 times higher in Asia than in Europe, and the observed ranges are in the range
of the emission flux ratios of the anthropogenic inventories (Figure 5 and Figure A2). In both anthropogenic inventories, the
ranges of the CO/BC ratio are similar between the two regions, which is also the case for the regional simulations (WRFchem–
FNL and WRFchem–ERA5) using the CAMS-GLOB-ANTv4.2 inventory. This suggests that this ratio is underestimated in
anthropogenic inventories in Asia. In addition, the large contribution of fire emissions in Asia compared to Europe is associated
with low values of this ratio, reinforcing the conclusion that the CO/BC ratio is underestimated in Asia (Figure A3).

3) For the HCHO/OA ratio, the observed value is similar in Europe and Asia ($\approx 0.3$). However, in both anthropogenic
inventories, the HCHO/OA ratio is much higher in Europe than in Asia, higher than 0.3 in Europe and lower than 0.1 in Asia
(Figure 5 and Figure A2). Moreover, fire emissions are associated with the low values of this ratio, and there are more fires
during the Asian campaign compared to the European one (Figure A3). In the regional simulations, the modeled ratio is 5
times higher in Europe than in Asia, suggesting that this ratio is overestimated in Europe and underestimated in Asia in both
anthropogenic inventories.

In summary, this section highlights significant differences in emission flux ratios in Europe compared to Asia, which are
consistent with the biases of the air quality model ensemble for the BC/OA, CO/BC and HCHO/OA ratios. The proportional
relationships between trace gases and BC/OA are not accurately replicated by the air quality model ensemble. These discrep-
ancies between the modeled and observed are related to emission inventories, as well as the representation of VOC and $O_3$
photochemistry in the models. Thus, the air quality model ensemble does not serve as a reliable tool for advancing understand-
ing in this context. However, the most relevant observed relationships could be valuable for constraining models and improving
emission inventories.





## 7 Conclusions and perspectives

This study presents a comprehensive investigation of the proportional relationships between carbonaceous aerosols and five satellite-observable trace gases, using data from the two EMeRGe campaigns conducted over Europe and Asia, with a specific focus on city plumes, along with identical instrumental payload. An air quality model ensemble, composed of two global models, CAMS–forecast and CAMchem–CESM2, and two regional WRFchem simulations with different meteorological input datasets, is constructed to represent the current state-of-the-art in atmospheric modeling.

The EMeRGe campaigns reveal relevant proportional relationships between carbonaceous aerosols and CO and HCHO. Observations show linear relationships between BC and OA, CO and BC, and HCHO and OA over a range of correlation coefficients (R) from 0.3 to 0.8. In city plumes, the relevant proportional relationships change because the R-values of BC and OA increase with $O_3$ (R $\approx$ 0.5). However, one-to-one analyses of these relationships are limited, and multivariate analyses are expected to yield higher levels of correlation.

The performance of the air quality model ensemble in reproducing the most relevant observed relationships is generally poor for both campaigns, with stronger or weaker linear relationships than observed. The modeled proportional relationships between carbonaceous aerosols and the five trace gases studied are almost linearly proportional for some trace gases, indicating large differences with observations. Furthermore, the correlations of the most relevant observed relationships are higher than those obtained for the modeled concentrations by the air quality model ensemble, reaching only R $\approx$ 0.5 for BC and R $\approx$ 0.3 for OA (Deroubaix et al., companion paper). This suggests that analysis of these proportional relationships is useful for improving air quality models, whose inaccuracies in replicating the most relevant observed relationships are related to inaccurate ratios in anthropogenic emission inventories.

The investigated hypothesis is that the proportional relationships between carbonaceous aerosols and trace gases are different in cities and their plumes compared to other environments. Based on this hypothesis, three questions are addressed:

***What are the relevant proportional relationships between carbonaceous aerosols and trace gases?***

Analyzing the proportional relationships between carbonaceous aerosols and the five trace gases for the most relevant linear relationships reveals that:

- the relative amount of BC to OA (*i.e.* the BC/OA ratio) is 3 times higher in the Asian campaign than in the European one, with higher linear relationship in Europe (R $\approx$ 0.8) than in Asia (R $\approx$ 0.6).

- the relative amount of CO to BC (*i.e.* the CO/BC ratio) is 1.5 higher in the Asian campaign than in the European one, with similar linear relationship for both campaigns (R $\approx$ 0.7).

- the relative amount of the HCHO to OA (*i.e.* the HCHO/OA ratio) is consistent in both campaigns, with varying linear relationship from low in Asia to high in Europe (R $\approx$ 0.3 compared to $\approx$ 0.7).

***Focusing on cities plumes, how are these relationships modified?***

The analysis of the most relevant proportional relationships observed in city plumes indicates that a significant part of the variability in carbonaceous aerosol concentrations is statistically related to the variability in $O_3$ concentrations. This result



shows that BC and OA concentrations are correlated with trace gases in city plumes, highlighting the potential of focusing on cities to obtain more information on aerosol composition.

### *Can these relationships be used to improve the satellite-based estimation of aerosols?*

Focusing on cities, this study implies that proportional relationships between carbonaceous aerosols and trace gases could be relevant for estimating carbonaceous aerosols from knowledge of trace gas composition. Despite the non-linearity of at-

mospheric processes and the complexity of VOC speciation and SOA formation, this study underscores the relevance of proportional relationships in city plumes, which could be used to infer information about the lower tropospheric concentration of carbonaceous aerosols from satellite retrievals of trace gases. Further studies investigating surface aerosol concentrations with aerosol optical depths, in combination with ground-based trace gas measurements and satellite trace gas retrievals, could lead to progress in answering this question.

*Author contributions.*  Conceptualization: AD, BG, PT, MV, JPB,

Data curation: MDAH, SB, BH, KK, FK, OOK, ML, KP, MP, HS, JS, BW,

Investigation: AD, MV, JPB,

Methodology: AD,

Resources: MV, BG, GB, GS,

Validation: AD, MV, BG, BS, KV,

Visualization: AD, IL,

Writing – original draft preparation: AD,

Writing – review & editing: All co-authors contributed.

*Competing interests.*  At least one of the (co-)authors is a member of the editorial board of Atmospheric Chemistry and Physics.

*Acknowledgements.*

Financial support:

Adrien Deroubaix acknowledge the European Union's Horizon 2020 research and innovation programme for supporting this work under the Marie Skłodowska-Curie grant agreement No 895803 (MACSECH — H2020-MSCA-IF-2019).

For the funding of the HALO aircraft and the contributions to the various missions via the German Research Foundation (DFG; HALO-

SPP 1294), the Max Planck Society (MPI), the Helmholtz-Gemeinschaft, and the Deutsches Zentrum für Luft- und Raumfahrt (DLR; all from Germany) are highly acknowledged.

This study was in part funded by the State and University of Bremen. The EMeRGe study in Bremen was funded by the DFG Project number 316834290, which is a sub project of the DFG SPP 1294: Atmospheric and Earth System Research with the "High Altitude and Long Range Research Aircraft" (HALO).



The scientific work of Flora Kluge, Klaus Pfeilsticker, and Benjamin Weyland has been supported by the German Research Foundation (DFG; grant nos. PF-384/7-1, PF384/9-1, PF-384/16-1, PF-384/17, PF-384/19, PF-384/24 and BU 2599/10-1).

       Johannes Schneider, Katharina Kaiser, and Stephan Borrmann acknowledge funding through the DFG (project no. 316589531).

Acknowledgements:

The computation of the simulations presented in this work was completed by different supercomputers:

   – For WRFchem, the authors gratefully acknowledge the resources granted by the Deutsches Klimarechenzentrum (DKRZ) granted by its Scientific Steering Committee (WLA) under project ID bb1260;

   – For WRF–CHIMERE, the authors gratefully acknowledge the support provided by Pablo Echevarria and the computing time allocated on Hypatia at IUP.


Data availability:

   – For the observational data, we thank the EMeRGe project for sharing the data, which are available through this website after registration: https://halo-db.pa.op.dlr.de/;

   – For CAMS–forecast, data are available through this website:
     https://ads.atmosphere.copernicus.eu/cdsapp#!/dataset/cams-global-atmospheric-composition-forecasts;

– For CAMchem–CESM2, data are available through this website: https://www.acom.ucar.edu/cam-chem/cam-chem.shtml.



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



# Appendix A:  Supplemental Material

**Table A1.** *Air quality model configurations: the model ensemble is composed of two global forecasts (CAMchem–CESM2 and CAMS–forecast) and two regional simulations using the WRFchem model.*

| Institution | NCAR | ECMWF | IUP |
|---|---|---|---|
| **Model** | CAMchem–CESM2.1 | IFS–CAMS | WRFchem (version 4.3.3) |
| ***Domain*** | | | |
| **Horizontal resolution** | 0.9 x 1.25° | 40 km | 10 km |
| **Vertical levels** | 56 | 137 | 37 |
| **Output frequency** | 6h | 3h | 1h |
| ***Emission*** | | | |
| **Anthropogenic** | CMIP6 | CAMS-GLOB-ANTv4.2 | CAMS-GLOB-ANTv4.2 |
| | (Feng et al., 2020) | (Granier et al., 2019) | (Granier et al., 2019) |
| **Biogenic** | MEGANv2.1 | MEGANv2.1 | MEGANv2.1 |
| | (Guenther et al., 2006) | (Guenther et al., 2006) | (Guenther et al., 2006) |
| **Fires** | QFED | CAMS-GFASv1.4 | FINNv1.5 |
| | (Darmenov et al., 2015) | (Inness et al., 2022) | (Wiedinmyer et al., 2011) |
| ***Gas and aerosol*** | | | |
| **Chemical mechanism** | MOZART4-T1 | CB05 | MOZART4-T1 |
| | (Emmons et al., 2020) | (Inness et al., 2019) | (Emmons et al., 2010) |
| **Aerosol scheme** | MAM4-VBS | IFS-AER | GOCART |
| | (Tilmes et al., 2019) | (Rémy et al., 2019) | (Chin et al., 2002) |
| **Boundary conditions** | none | none | CAMchem–CESM2.1 |





**Table A2.** *Name and location of the emission point of numerical gaseous tracers from selected city. For Paris and London, five locations are used.*

| Country | City | Longitude | Latitude |
|---|---|---|---|
| **EMeRGe - Europe** | | | |
| France | Paris C | 2.34 | 48.85 |
| France | Paris N | 2.34 | 48.95 |
| France | Paris S | 2.34 | 48.75 |
| France | Paris W | 2.24 | 48.85 |
| France | Paris E | 2.44 | 48.85 |
| France | Marseille | 5.37 | 43.32 |
| France | Le Havre | 0.15 | 49.47 |
| France | Lyon | 4.84 | 45.75 |
| United Kingdom | London C | -0.12 | 51.50 |
| United Kingdom | London N | -0.12 | 51.60 |
| United Kingdom | London S | -0.12 | 51.40 |
| United Kingdom | London W | -0.22 | 51.50 |
| United Kingdom | London E | -0.02 | 51.50 |
| Germany | Manchester | -2.23 | 53.46 |
| Germany | Munich | 11.58 | 48.13 |
| Germany | Cologne | 6.96 | 50.93 |
| Germany | Stuttgart | 9.17 | 48.77 |
| Italy | Milan | 9.18 | 45.46 |
| Italy | Genoa | 8.87 | 44.41 |
| Italy | Turin | 7.66 | 45.07 |
| Italy | Venice | 12.25 | 45.44 |
| Italy | Rome | 12.47 | 41.89 |
| Spain | Barcelona | 2.16 | 41.38 |
| Belgium | Brussels | 4.35 | 50.84 |
| Netherlanders | Rotterdam | 4.47 | 51.90 |



**Table A3.** *Name and location of the emission point of numerical gaseous tracers from selected city. For Taipei and Manila, five locations are used.*

| Country | City | Longitude | Latitude |
|---|---|---|---|
| **EMeRGe - Asia** | | | |
| Taiwan | Taipei | 121.50 | 25.06 |
| Taiwan | Taoyuan | 121.22 | 24.95 |
| Taiwan | Taichung | 120.68 | 24.16 |
| Taiwan | Tainan | 120.20 | 22.99 |
| Taiwan | Kaohsiung | 120.35 | 22.62 |
| Philippines | Manila C | 121.03 | 14.60 |
| Philippines | Manila W | 120.93 | 14.60 |
| Philippines | Manila N | 121.03 | 14.70 |
| Philippines | Manila E | 121.13 | 14.60 |
| Philippines | Manila S | 121.03 | 14.50 |
| China | Guangzhou | 113.29 | 23.11 |
| China | HongKong | 114.13 | 22.35 |
| China | Xiamen | 118.15 | 24.48 |
| China | Fuzhou | 119.33 | 26.04 |
| China | Wenzhou | 120.70 | 28.00 |
| China | Hangzhou | 120.18 | 30.28 |
| China | Shanghai | 121.48 | 31.18 |
| China | Nantong | 120.89 | 31.99 |
| Japon | Osaka | 135.50 | 34.65 |



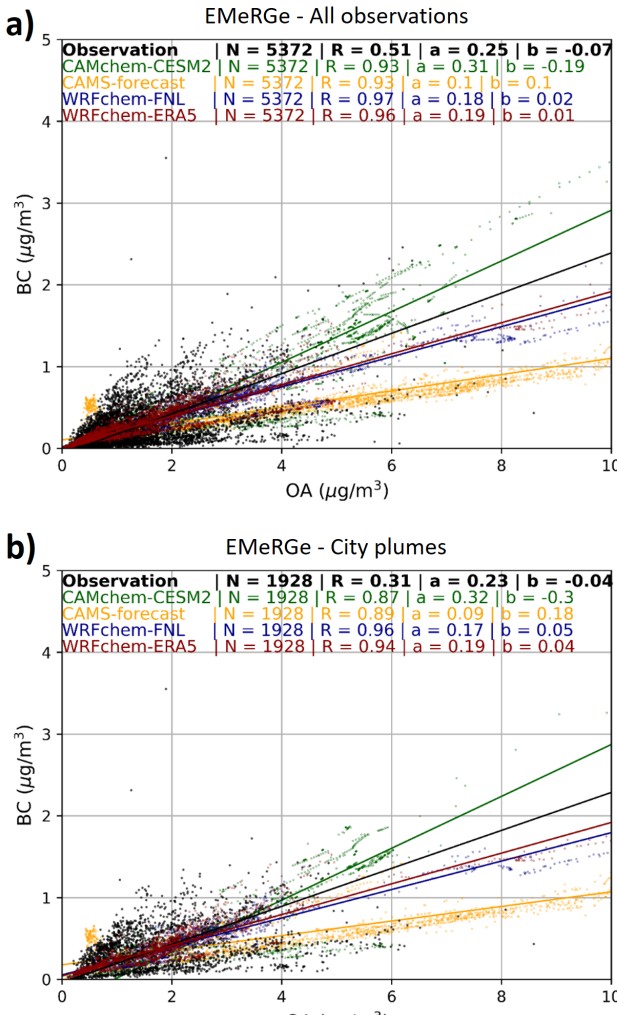

**Figure A1.** *Comparison of black carbon concentrations (BC in μg/m³) with organic aerosol (OA in μg/m³) during the two EMeRGe campaigns: (a) for all observations, and (b) for the observations which identified in city plumes. Scatter plots of the concentrations of BC against OA for the observations and for an air quality model ensemble composed of two global simulations, CAMchem–CESM2 (green line) and CAMS–forecast (orange line) and two regional simulations, WRFchem–FNL (blue line) and WRFchem–ERA5 (red line). Statistics of linear regression analysis (using reduced major axis regression) are given at the top for each simulation.*





**Figure A2.** *Maps of emission flux ratios from the anthropogenic inventory EDGARv4.3.2 (sum of all sectors): a) for BC/OA in Europe for July 2017, b) for BC/OA in Asia for April 2018, c) for CO/BC in Europe for July 2017, d) for CO/BC in Asia for April 2018, e) for HCHO/OA in Europe for July 2017, and f) for HCHO/OA in Asia for April 2018.*





**Figure A3.** *Maps of emission flux ratios from the fire inventory GFED4 (sum of fire types): a) for BC/OA in Europe for July 2017, b) for BC/OA in Asia for April 2018, c) for CO/BC in Europe for July 2017, d) for CO/BC in Asia for April 2018, e) for HCHO/OA in Europe for July 2017, and f) for HCHO/OA in Asia for April 2018.*