# Peer review of "Proportional relationships between carbonaceous aerosols and trace gases in city plumes of Europe and East Asia"

_EGUsphere, 2024_

## Referee Comment (RC1)

Summary:

This paper studied the correlation relationship between black carbon (BC) and organic aerosol (OA), and the correlations of two types of carbonaceous particles with other gaseous pollutants in both airborne observations and associated four different modeling frameworks. The airborne platforms that flew over Europe and East Asia were equipped with identical instrument payloads, which made the concentrations of different types of pollutants comparable. The flights encompass several major urban plumes, making it possible to further study the in-plume relationships of the related pollutants.

This paper presented comprehensive analyses of all detected pollutants and quantitatively analyzed the correlations and relative abundance of each atmospheric species. Observations in Europe and Asia showed variated pollutants abundances. Different levels of correlation between gaseous pollutants and particulate matters reflect dynamic natures of co-emission intensities. By comparing the observations with modeling outputs, the study found that stronger correlations between primary gaseous pollutants and particulate matters were widely presented in modeling than observations. In contrast, the spatial variance of the observed relative abundance of certain gaseous species and BC or OA in two city plumes were not reproduced by modeling. This discrepancy is partially attributed to the inaccurate ratios in anthropogenic emission inventories. The paper suggests more observational constraints including satellite-, airborne-, and ground-based measurements were incorporated into the compilation of the inventories.

General suggestions:

1. Figure 1: This figure mainly illustrates the flight trajectories of two campaigns and the portions of urban plume transects. Were those urban plume transects discretely distributed along the trajectories like it shows in the figure, or should be temporally and spatially continuous during the campaign. Given the vertical profiles of the sampled pollutants are not the main topic to be discussed in this study, a 2-D figure with trajectory lines overlaid on it should be sufficient to illustrate the idea. The urban plume portion can be highlighted in a different color. 2-D map would also help to visualize the boarders and populous regions.
2. Figure 2: This figure mainly shows the strong correlation between BC and OA among almost all frameworks. The visualization of this figure can be further improved. Now there is a high degree of data overlap in both panels. The color of data points and relative size of markers make the data difficult to distinguish. May consider present the correlation in each framework (observations, each modeling settings) in a separate subplot, and use the scatter density plot to illustrate the correlations.
3. Line 181: The higher BC/OA ratio in Asia is attributed to wildfires in Indochina. Given wildfires contribute a large amount of primary OA and secondary OA precursors as well, is there any chemical analysis of OA during the campaign (or source apportionment) to indicate the contributions of possible OA sources? HCHO was regarded as a better proxy for OA according to the correlation analysis. Any observations from the mass spectrometers can be added to infer the possible mechanisms behind this correlation?

4. Figure 3: In panel b, at high BC concentrations, concentrations of SO2, HCHO, and NO2 tend to be constant as BC increases. Any interpretation of this relationship here? Similar patterns were found in Figure 4b.
5. Table 3. Any explanation on zero R-value of CAMS-forecast O3 to OA in Europe?
6. Secondary OA takes up a large portion of mass in total OA. I suppose the ratios of emission fluxes in Figure 5 mainly quantifies the relative abundance of primary sources (BC, POA, etc.), but not considering the SOA precursors that may further form particle-phase OA. The SOA formation mechanisms depend on model selections and need to be considered in inventory evaluation. In addition, a positive matrix factorization (PMF) analysis on observed chemical composition of aerosol can also help to provide source apportionment information of OA.

**Line-by-line suggestions:**

1. Line-230: "R-value"
2. Line-264 to 267: These two sentences are a bit redundant in expression.

---

## Author Comment (AC1)

**Revision egusphere-2024-521**

**To editor**

Dear Editor,

We thank the reviewers for their constructive comments, which have greatly helped to refine the study of the main results. We now present a detailed analysis of observed linear and non-linear gas-aerosol relationships, and we discuss our results with the perspective of estimating aerosol concentrations using satellite retrievals of trace gases. In this new version, we redefine the focus of our study on the observed gas-aerosol relationships during the EMeRGe aircraft campaigns, thus removing the analysis of air quality models (which has been addressed by the revision of the article egusphere-2024-516).

The reviewers' comments showed us that our results on the analysis of observed gas-aerosol relationships constitute the most essential part of the analysis of the first version dealing with both observations and models. The analysis of observed gas-aerosol relationships deserves to be conducted independently, as these relationships have received very little attention in the literature. The analysis of gas-aerosol relationships in air quality model simulations would be more relevant in a dedicated article to follow. Therefore, we understand your decision to reject our other article (egusphere-2024-516) that focuses on these relationships in models and observations. However, we believe that analysis of modeled gas-aerosol relationships is an important aspect of understanding and improving models. Our responses to the reviewers' comments along these lines are highlighted in purple.

We are pleased to present this revised version, which represents an important exploratory step to show that it is possible to estimate carbonaceous aerosol using gas-aerosol relationships in the atmosphere with aircraft measurements, before investigating the use of trace gas satellite retrievals to estimate carbonaceous aerosol. The novelty of this revision is to compare the performance of multi-linear regression with machine learning, which could account for non-linear atmospheric processes. Furthermore, we show that the limitations of multi-linear regression are related to the different types of pollution that can be distinguished by trace gas ratios. Our results suggest that satellite trace gas data could be a valuable tool for estimating BC and OA concentrations using a machine learning approach trained on ground-based measurements, especially over megacities with high and localized emissions.

In the new version of the manuscript, we have changed the title, abstract, introduction, structure, and conclusions of the article accordingly (see our responses to the reviewers' comments below). We have also taken care to improve the readability of the manuscript and figures. Please find enclosed our point-by-point responses to the reviewers' comments, as well as a separate document with all corrections to the manuscript, highlighted in blue for additions and red for removals.

We hope you will consider this revised manuscript for publication in ACP.

Yours sincerely
Adrien Deroubaix, on behalf of all co-authors

**Responses to the report of referee #1**

Summary: This paper studied the correlation relationship between black carbon (BC) and organic aerosol (OA), and the correlations of two types of carbonaceous particles with other gaseous pollutants in both airborne observations and associated four different modeling frameworks. The airborne platforms that flew over Europe and East Asia were equipped with identical instrument payloads, which made the concentrations of different types of pollutants comparable. The flights encompass several major urban plumes, making it possible to further study the in-plume relationships of the related pollutants.

This paper presented comprehensive analyses of all detected pollutants and quantitatively analyzed the correlations and relative abundance of each atmospheric species. Observations in Europe and Asia showed variated pollutants abundances. Different levels of correlation between gaseous pollutants and particulate matters reflect dynamic natures of co-emission intensities. By comparing the observations with modeling outputs, the study found that stronger correlations between primary gaseous pollutants and particulate matters were widely presented in modeling than observations. In contrast, the spatial variance of the observed relative abundance of certain gaseous species and BC or OA in two city plumes were not reproduced by modeling. This discrepancy is partially attributed to the inaccurate ratios in anthropogenic emission inventories. The paper suggests more observational constraints including satellite-, airborne-, and ground-based measurements were incorporated into the compilation of the inventories.

We would like to thank the reviewers for their constructive comments. In the revised version, we have made a number of changes in order to focus on the analysis of the essential results of observed gas-aerosol relationships. As we have refocused the study on the analysis of observed pollutant concentrations only, we have decided to delete the results on modeled pollutant concentrations.

In the new version, emission inventories are out of scope. The previous conclusions on the possibility of constraining emission inventories have been removed because another article would

be needed to thoroughly analyze the biases in the gas/aerosol ratios at emission with more observational constraints, including satellite, airborne, and ground-based measurements. The revised version presents an in-depth analysis of the gas/aerosol relationships observed during the two campaigns, which is a necessary study prior to model analysis. We investigate linear and non-linear relationships using linear regression and machine learning approaches.

The results of the revised version show that aircraft measurements of trace gas concentrations can be used to estimate aerosol composition, highlighting the links between carbonaceous aerosols and trace gases. This is an important step towards the overarching goal of the article, which is to estimate aerosol composition using trace gas satellite retrievals.

**General suggestions:**

1. Figure 1: This figure mainly illustrates the flight trajectories of two campaigns and the portions of urban plume transects. Were those urban plume transects discretely distributed along the trajectories like it shows in the figure, or should be temporally and spatially continuous during the campaign. Given the vertical profiles of the sampled pollutants are not the main topic to be discussed in this study, a 2-D figure with trajectory lines overlaid on it should be sufficient to illustrate the idea. The urban plume portion can be highlighted in a different color. 2-D map would also help to visualize the borders and populous regions.

We agree with the reviewer that this figure could be improved. However, the former Figure 1 showing all individual flight paths is not essential since our analyses always consider all flights from a region together (either the complete flights or the flight legs sampling city plumes). Therefore, this figure is now in the Supplement to illustrate the areas where the campaign was conducted and the parts where city plumes were identified (new Figure A1).

2. Figure 2: This figure mainly shows the strong correlation between BC and OA among almost all frameworks. The visualization of this figure can be further improved. Now there is a high degree of data overlap in both panels. The color of data points and relative size of markers make the data difficult to distinguish. May consider present the correlation in each framework (observations, each modeling settings) in a separate subplot, and use the scatter density plot to illustrate the correlations.

This figure was difficult to read. Since we have removed model simulations from our analysis, the figure has been simplified by showing observations from both campaigns. We have adjusted the point size and transparency to improve readability (see new Figure 3).

We thank the reviewer for the suggestion to use scatter density plots, and we have used them to illustrate the agreement between our statistical models of carbonaceous aerosol based on trace gas concentrations in new Section 4 (new Figures 4 and 5).

3. Line 181: The higher BC/OA ratio in Asia is attributed to wildfires in Indochina. Given wildfires contribute a large amount of primary OA and secondary OA precursors as well, is there any

chemical analysis of OA during the campaign (or source apportionment) to indicate the contributions of possible OA sources? HCHO was regarded as a better proxy for OA according to the correlation analysis. Any observations from the mass spectrometers can be added to infer the possible mechanisms behind this correlation?

The reviewer correctly points out the need for further analysis to understand the contribution of fire emissions to the observed pollutant concentrations and ratios. The attribution of the higher BC/OA ratio in Asia to wildfires was not correct. Although it may be part of the answer, we cannot confirm this hypothesis based on the analyses presented. There are many other potential contributing factors, which are now discussed in Section 3.3 (L 216-220 in the new version).

*"The low R²-value obtained for the Asian campaign suggests that a larger variability of pollution types in terms of BC/OA ratio has been sampled in Asia. Nevertheless, it seems that there are different branches for the BC-OA relationship for both campaigns (Figure 3). This result could be linked to different types of combustion associated with different BC/OA ratios, which are associated to higher values in fire emission inventory (e.g., Giglio et al., 2013) than in anthropogenic emission inventory (e.g., Crippa et al., 2018). "*

We have extended the analysis of the HCHO-OA relationship in the revised version, showing that this relationship is strong in the European campaign but not in the Asian campaign (Section 3.2). The strong differences between regions are probably related to a larger variability in the types of pollution sampled during the Asian campaign (L 186-188 in the new version):

*"Liao et al (2019) showed that the HCHO-OA relationship is strong, reaching R²-values up to 0.35, and that the HCHO/OA ratio has different values depending on the environment. We note that the HCHO-OA relationship is strong in Europe, so that HCHO is as relevant a proxy for OA as CO, whereas this is not the case in Asia (R² = 0.44 in Europe vs. 0.07 in Asia). These results further suggest that the Asian campaign sampled more diverse types of pollution than the European campaign."*

In addition, we need to study the sum of OA because secondary formation is associated with gas precursors of O3, such as NO2 and HCHO. Chemical analysis of OA or source attribution is possible with the EMeRGe datasets. Since this kind of analysis must be performed for individual flights on a case study basis, it is beyond the scope of this article, which focuses on common gas-aerosol relationships in the atmosphere and their interest in estimating aerosol composition from trace gas concentrations.

4. Figure 3: In panel b, at high BC concentrations, concentrations of SO2, HCHO, and NO2 tend to be constant as BC increases. Any interpretation of this relationship here? Similar patterns were found in Figure 4b.

We thank the reviewer for this comment, which contributed to an important addition in the revised version, focusing on the fact that the scatter plots of trace gas aerosol concentrations show a separation into several "branches". We see that the slopes are positive, but the range of SO2, HCHO, and NO2 concentrations is lower than the range of O3 and CO concentrations because

of their shorter lifetimes. Some trace gas concentrations tend to be constant, but only for certain cases (i.e. one of the "branches"). We investigate the causes of these "branches" in the new Section 4 with four trace gas ratios known to be relevant for air pollution analysis (L 259-264 in the new version):

*"To analyze the types of pollution associated with the two branches, we use four ratios of the five trace gases studied, (i) SO2/NO2 and (ii) CO/NO2, which are representative of combustion types, as well as (iii) HCHO/NO2 and (iv) O3/NO2, which are representative of oxidation regime and activity. The SO2/NO2 and CO/NO2 ratios are associated with different values for anthropogenic or fire emissions due to different combustion efficiencies and fuels (e.g., Tang et al., 2019; Lama et al., 2020), whereas the HCHO/NO2 and O3/NO2 ratios depend on the level of O3 precursors and photochemical activity (e.g., Zhang et al., 2009; Souri et al., 2023)."*

Furthermore, the figures have been revised to put trace gases (i.e. the predictors) on the x-axis and the aerosol (i.e. the predicted variables) on the y-axis, which improves their readability (new Figures 1, 2 and 3).

5. Table 3. Any explanation on zero R-value of CAMS-forecast O3 to OA in Europe?

It is expected that there will be some variability between model simulations and that there may be strong outliers in some situations. For this reason, we used several models in our previous version study. However, all results on model simulations have been removed from this revised version to focus on the observed gas-aerosol relationships, as the analysis of air quality models would deserve a dedicated article.

6. Secondary OA takes up a large portion of mass in total OA. I suppose the ratios of emission fluxes in Figure 5 mainly quantifies the relative abundance of primary sources (BC, POA, etc.), but not considering the SOA precursors that may further form particle phase OA. The SOA formation mechanisms depend on model selections and need to be considered in inventory evaluation. In addition, a positive matrix factorization (PMF) analysis on observed chemical composition of aerosol can also help to provide source apportionment information of OA.

The model simulations and emission inventories have been removed from the revised version and are therefore outside the scope of the new version. However, the analysis of the results of the model simulations and emission inventories should be addressed in a dedicated article.

It is important to emphasize that SOA can contribute to a large fraction of the total OA.This is a crucial aspect for understanding the relationships between trace gases and OA, which is now mentioned in Section 3.2 (L 198-200 in the new version) and in Section 4.2 (L 298-300 in the new version):

*"The strong O3-OA relationship in city plumes may be related to the co-emission of BC, OA and CO along with O3 precursors and SOA formation (e.g., Turpin and Huntzicker, 1991; Jimenez et al., 2009; McMeeking et al., 2011; Zhang et al., 2015; Yoon et al., 2021)."*

*"The large variety of factors involved in the formation of SOA, which could account for a significant proportion of OA in city plumes (Freney et al., 2014), is an important limitation for the estimation of OA using MLR."*

**Line-by-line suggestions:**

1. Line-230: "R-value"

The corresponding sentence has been removed from the revised version.

2. Line-264 to 267: These two sentences are a bit redundant in expression.

This repetition has been deleted.

**Responses to the report of referee #2**

Summary:

The paper by Deroubaix et al. investigates the proportional relationships between carbonaceous aerosols and trace gases using measurements from the EMeRGe aircraft campaigns over Europe and East Asia. The authors also evaluate the performance of regional and global models in simulating these relationships. The aircraft measurements and model evaluations provide valuable insights for the estimation of carbonaceous aerosols from the trace gases. Overall, there are potentially some interesting aspects of this paper, but at the same time there are a large number of results that are not explained or justified. The paper does not meet the publication criteria of ACP, and I cannot recommend it for publication.

We understand the criticism of reviewer 2 and thank them for their constructive comments. The presented study is part of a larger project investigating gas-aerosol relationships (funded by the European Commission through a Marie Skłodowska-Curie grant awarded to projects proposing an innovative idea). The goal of the project is to determine the potential of estimating the aerosol composition from the trace gas composition observed by satellites. The reviewers' comments showed us that it was a mistake not to refer to the overarching goal of the article, as it is necessary to justify the manuscript's focus on megacities and specific trace gases, which is now explained (L 42-48 in the new version):

*"The processes leading to aerosol production and loss are complex on small scales and can be highly non-linear. Moreover, observations at these small scales are very limited. At larger scales, the relative amounts of aerosols and trace gases may lie within a narrow range of variability.*

*Therefore, small-scale non-linearities can be statistically smoothed at larger scales. Here, we aim to identify relevant gas-aerosol relationships, whether linear or non-linear, and to determine the accuracy of aerosol composition estimates provided by statistical models exploiting these relationships. This new approach can shed light on aerosol composition, especially in megacities, where gas and aerosol emissions are spatially localized and the day-to-day variability is low."*

We agree and apologize that some of the results presented in the first submission were not discussed in sufficient detail because the aims and scientific questions of the article were too broad and ambitious to be achieved. We have now clarified the focus on the analysis of observed concentrations only in the context of the overarching goal of the article, and adapted the structure of the revised version accordingly. We have therefore removed the results on modeled concentrations. Instead, the revised version presents an in-depth analysis of the gas-aerosol relationships observed during the two campaigns. We investigate both linear and non-linear relationships using linear regression and machine learning approaches. Our results represent an important step towards the goal of the project. The manuscript has been substantially revised to reflect this clearer focus, including the introduction, results presentation, and discussion.

**Major comments:**

1、 In the observational analysis, the authors primarily present their observational results without providing any physical explanations or comprehensive analysis. As a result, the paper reads more like an experimental report. The entire paper lacks thorough discussions and comparisons with relevant research. For instance, only three references are cited in Sections 3-7. The authors should compare the results with the previous studies and offer new insights to improve the scientific quality of this paper.

We agree with the reviewer that the references and comparisons with relevant research needed to be improved from the first version. For this revised version, we have tried to exhaustively cite relevant references to show the state of knowledge on gas-aerosol relationships and to support our results. Although few articles have investigated gas-aerosol relationships, we now discuss the existing literature from observations at ground-based stations in urban areas and remote locations, from aircraft measurements, and from satellite retrievals, for BC-CO (e.g, Baumgardner et al., 2002; Pan et al., 2011; Spackman et al., 2008, Arellano et al., 2010; Mok et al., 2017), for OA-BC (e.g., Turpin and Huntzicker, 1991; Novakov et al., 1997; Sahu et al., 2011; Becerril-Valle et al., 2017), and for OA-HCHO (e.g., Liao et al., 2019).

In addition, we highlight the relevance of the EMeRGe datasets, which consist of two large and contrasting datasets of airborne observations from different regions and seasons, for our study of the relevant statistical relationships between carbonaceous aerosols and trace gases (L 68-76 in the new version):

*"The advantage of aircraft is that they sample different environments (in space), but a disadvantage is that this information is provided as a snapshot (in time). The two EMeRGe (Effect*

*of Megacities on the Transport and Transformation of Pollutants on the Regional to Global Scales) air campaigns, which took place in Europe in 2017 and East Asia in 2018, are particularly interesting for studying the relationships between carbonaceous aerosols and trace gases. This is because the German research aircraft, called HALO (High Altitude and LOng Range Research Aircraft), carried an identical instrumental payload for both campaigns, with flight plans dedicated to studying the compositions of city plumes (Andrés Hernández et al., 2022; Förster et al., 2023; Lin et al., 2023). As background and city plume concentrations were different during the European and Asian campaigns (Förster et al., 2023), the aerosol and trace gas measurements made offer a unique opportunity to analyze the statistical relationships between carbonaceous aerosol and trace gas compositions in city plumes in the two regions."*

2、 In the model evaluation, the authors have performed only a basic comparison of model and measurements, and some logic seems misleading. The model bias in the proportional relationships should result from a combination of biases in both aerosols and trace gases. The modeled aerosol concentrations, especially in the source regions, are mainly determined by emission, transport, and particle chemistry and physical processes. The interpretations of the model bias in modeling the proportional relationships and their relation to the emission inventory are not sufficiently justified. The authors should first prove that whether the model bias in simulating proportional relationships primarily arises from the emission inventory (or model processes).

The results of the model simulations have been removed from the revised version, which now presents an in-depth analysis of the observed gas-aerosol concentrations. As mentioned by the reviewer, the interpretation of model bias in the previous version was not sufficient to support our conclusion on emission inventories. However, we believe that a dedicated article is needed to understand the response of air quality models in both regions.

3、 The authors proposed three major scientific questions. The analysis of city plumes is critical but too brief. Additionally, the authors barely discussed the satellite-based estimation of aerosols. On the contrary, they extensively presented model results, yet they failed to integrate model evaluations into the major scientific questions.

We thank the reviewer for this comment and agree that it is crucial to clearly redefine the goals and scope of the study. We now present two precise research questions that are explored and answered in this paper, as well as a clear explanation of the overarching goal of the study (which in itself goes beyond this study, but this study lays the basis for further research in this perspective). The new research questions are (L 80-81 in the new version):

*"What are the most relevant proxies to estimate carbonaceous aerosols using the five trace gases studied?*

*What is the performance of a machine learning method to estimate carbonaceous aerosols using the five trace gases studied?"*

The revised version presents a comprehensive analysis of these questions and concludes with a discussion of the potential of gas-aerosol relationships to estimate aerosols from satellite observations of trace gases, which represents the overarching goal.

4、 The organization of the paper is poor. The three scientific questions have been buried in Sections 3-7, making it difficult to hold the interest of readers. I recommend that the paper should be reorganized around the scientific questions.

Following the recommendation of the reviewer, the manuscript has been reorganized according to the revised scientific questions. For further details, please refer to the response to comment #3.

**Specific comments (in chronological order)**:

1、 Lines 87-89: The paper focuses on the observed proportional relationships between aerosols and trace gases, as well as the model performance in simulating these relationships. The introduction (Lines 53-87) mainly emphasizes the significance of aircraft-based and satellite-based measurements for aerosols and trace gases. The authors should concentrate more on relevant observations and model simulations about the proportional relationship between aerosols and trace gases.

The introduction has been rewritten with a clearer description of the overarching goal of our study and with new specific research questions (see Major comment #3).

2、 Line 94: "cities plumes" -> "city plumes"; also check elsewhere

Throughout the document, this has been corrected and verified.

3、 Lines 100-104: The structure of the paper seems like a mechanical repetition and requires reorganization (see Major comments #4).

We have reorganized the manuscript and revised this paragraph to adequately explain the new structure. This paragraph has changed as follows (L 82-87 in the new version):

*"The two EMeRGe aircraft campaigns and the identification of city plumes are presented in Section 2. First, the most relevant gas-aerosol relationships are analyzed based on linear regression analysis (Section 3). Second, the shared gas-aerosol relationships between the two campaigns are analyzed by comparing statistical methods based on either multi-linear or non-linear relationships using a machine learning method (Section 4). Finally, the relevance of the relationships between carbonaceous aerosols and trace gases is discussed in the perspective of estimating aerosol composition based on the trace gas composition observed by satellites (Section 5)."*

4、 Figure 2 : How can we ensure consistency in time and space between aircraft observations and models with different resolutions? Additionally, the scatter points in the figure overlap with each other. All figures (including Figures 3-5) should be replotted to make them more clear.

The results of model simulations have been removed from the revised version, so an explanation of the temporal and spatial consistency between aircraft observations and models is no longer relevant.

Nevertheless, in the previous version, to ensure consistency in time and space between aircraft observations and models of different resolutions, we interpolated the concentrations of each model along the flight path (x, y z, t) with a triple interpolation (bilinear horizontally, linear vertically, and linear temporally between two time steps), as is standard for comparing aircraft measurements with air quality  models (e.g. Menut et al., 2015; Petetin et al., 2015; Deroubaix et al., 2018 and 2019; Flamant et al., 2018). It should be noted that the time-averaging of observations corresponds to measurements taken from a moving aircraft. In other words, an averaging time step of 1 min corresponds to a horizontal distance of around 10 km, and an averaging time step of 10 min corresponds to a horizontal distance of around 100 km, which is comparable to the spatial resolution of regional and even global models.

All figures have been modified to improve readability by adjusting point size and transparency of the scatter plots to account for point overlap (see new Figures 1, 2, and 3).

5、 Lines 168-169 and Lines 172- 173: The author has not provided physical explanations for the higher R-value in Europe or for the much higher BC/OA ratio in Asia. Although the author mentioned wildfires in Indochina in Line 182, the explanation is not convincing and requires more comparisons with previous studies.

We recognize that physical explanations to interpret the higher correlation in Europe and higher ratio in Asia were lacking. The attribution of the higher BC/OA ratio in Asia to wildfires was not correct. Although it may be part of the answer, we cannot confirm this hypothesis based on the analyses presented. There are many other potentially contributing factors which are now discussed in Section 3.3 (L 216-220 in the new version).

*"The low R²-value obtained for the Asian campaign suggests that a larger variability of pollution types in terms of BC/OA ratio has been sampled in Asia. Nevertheless, it seems that there are different branches for the BC-OA relationship for both campaigns (Figure 3). This result could be linked to different types of combustion associated with different BC/OA ratios, which are associated to higher values in fire emission inventory (e.g., Giglio et al., 2013) than in anthropogenic emission inventory (e.g., Crippa et al., 2018). "*

In the revised version, we show that these differences are likely due to a larger variability of pollution types sampled during the Asian campaign compared to the European campaign. This is now thoroughly explained in the manuscript, since the abstract (L 10-12 in the new version),

section 3.1 (L 159-161), section 3.2 (L 178-180), section 3.3 (L 216-220) and in the conclusion (L 332-334), which is:

*"In addition, the strength of the gas-aerosol relationships is much stronger during the European campaign than during the Asian one in terms of R²-values, suggesting that the variability of the pollution sampled during the European campaign was lower than during the Asian one."*

Overall, Section 3 shows that there are differences between the two campaigns for the relevant gas-aerosol relationships in terms of strength (i.e. R²-values) and gas-aerosol ratios (i.e. the slopes). Despite these differences, statistical approaches based on the five trace gases studied as predictors show higher correlation than the one by one gas-aerosol relationships in each region (Section 4). Using machine learning, R² reaches > 0.9, while the best trace gas proxy (which is CO as a proxy for BC in Europe) was only about 0.7.

6、 Lines 179-181: Why do all the models simulate higher R-values than observations ? Please explain.

The results of model simulations have been removed from the revised version. This important aspect of the strength of gas-aerosol relationships is no longer within the scope of the article, and we believe it would require a dedicated article.

7、 Section 3.2, Section 4.2, and Section 5.2: The analysis of urban plumes is crucial but too brief. Also, suggest merging these sections.

We thank the reviewer for this comment and agree that the analysis of the city plumes is crucial. The gas-aerosol relationships observed during both campaigns are now examined and discussed for each campaign (along with all observations) throughout the manuscript. This was missing in the previous version.

8、 Figure A1: The city plumes of Asia and Europe come from different environments, so can they be plotted together in Figure A1? Can the models well reproduce the city plumes? Additionally, how can we ensure consistency in time and space between observations and models? Please explain.

We agree that separate graphs should be analyzed for the Asian and European campaigns before merging the datasets. BC-OA relationships in city plumes are now presented for each campaign in the new Figure 3.

The results of model simulations have been removed from the revised version. This important aspect of modeled city plumes is no longer within the scope of the article and would require a dedicated article.

Regarding temporal and spatial consistency between aircraft observations and models, see the response to specific comment #4.

9、 Lines 189-191: In the city plumes, why does the observed R-value decrease in observations but hardly decrease in the models? Please explain.

A possible explanation for the observed decrease in R-value that is not seen in the models is that the variability of the BC/OA ratio at emission in urbanized areas is underestimated in the inventories. However, the results of the model simulations have been removed from the revised version, which would require a dedicated article to analyze them in depth.

10、 Lines 209-213 and Table 1: Why are the R-values of the observed BC with the trace gases (CO, HCHO, $NO_2$, and $SO_2$) higher in Europe than those in Asia? Why is the R-value of the observed BC with $O_3$ lower in Europe than that in Asia? Why are the R-values of the observed BC with the three trace gases ($NO_2$, $O_3$, and $SO_2$) lower than the R-values of BC with the two trace gases (CO and HCHO) for the two campaigns? It requires more comparisons with previous studies and discussions.

These are interesting points to understand the observations. In the revised version, we further analyze the differences observed in the two campaigns and compare the relevant relationships with the literature (BC-CO, OA-BC and OA-HCHO). Some gas-aerosol relationships are observed with high $R^2$-values only because they share emission sources. In the conclusion of section 3, we summarize the interpretation of the differences between the two campaigns (L 221-228 in the new version):

*"In conclusion of Section 3, the best proxy for the two carbonaceous aerosols is CO during both campaigns. Focusing on city plumes, O3 also appears to be a good proxy, almost as relevant as CO, suggesting that in air masses from city plumes have active chemistry because BC, OA and CO are co-emitted with precursors of O3 formation. The level of correlation of carbonaceous aerosols with the five trace gases studied is higher during the European campaign than during the Asian campaign, which suggests that more consistent pollution types were sampled in Europe. In addition, the gas-aerosol relationships studied seem to have several branches for both campaigns, as well as the BC-OA relationship, which may be related to different combustion types or oxidation regimes, and which shows the limitations of a carbonaceous aerosol proxy based on linear regression."*

After assessing the differences between the two campaigns, we examine in more detail the different types of pollution sampled during the two campaigns, using four trace gas ratios known for their relevance in air pollution analysis (L 259-264 in the new version):

*"To analyze the types of pollution associated with the two branches, we use four ratios of the five trace gases studied, (i) SO2/NO2 and (ii) CO/NO2, which are representative of combustion types, as well as (iii) HCHO/NO2 and (iv) O3/NO2, which are representative of oxidation regime and activity. The SO2/NO2 and CO/NO2 ratios are associated with different values for anthropogenic or fire emissions due to different combustion efficiencies and fuels (e.g., Tang et al., 2019; Lama et al., 2020), whereas the HCHO/NO2 and O3/NO2 ratios depend on the level of O3 precursors and photochemical activity (e.g., Zhang et al., 2009; Souri et al., 2023)."*

11、 Line 216: The CO/BC ratio is 168.01 ppb per µg/m$^3$ of BC in Europe and 243.75 ppb per µg/m$^3$ of BC in Asia?

The reviewer is right. The values should have been 168.01 ppb per µg/m3 BC in Europe and 243.75 ppb per µg/m3 BC in Asia in Table 1 of the first version.

In the revised versions, we present the relationship as BC-CO instead of CO-BC, in order to place trace gases (i.e. predictors) on the x-axis and aerosols (i.e. predicted variable) on the y-axis. Values have been modified accordingly in the manuscript, which is more in line with the literature (e.g. Baumgardner et al., 2002; Pan et al., 2011; Spackman et al., 2008).

12、 Lines 216-219: Could you provide any comments on the differences in observed CO/BC and HCHO/BC ratios between Europe and Asia?

We have added a detailed discussion of the differences between the two campaigns, as already mentioned in specific comment #10.

13、 Table 1 : Could you provide any comments on the zero R-value of BC with O$_3$ in the model (e.g., WRFchem–FNL and WRFchem–ERA5)? Why are the modeled R-values between BC and O$_3$ by CAMS–forecast negative in Europe but positive in Asia? Please explain.

The results of model simulations have been removed from the revised version. This comment emphasizes the importance of using several models (i.e. an ensemble) to analyze variability between models, as their results can be very different. This important aspect no longer falls within the scope of the article and would require a dedicated article.

14、 Lines 222-223: Why is the observed CO/BC ratio in Europe much lower than that in Asia? Additionally, why is the CO/BC ratio in Europe modeled by CAMchem-CESM2 twice as high as that in Asia? Please explain.

15、 Lines 224-225 and Lines 232-233: Why do models typically overestimate the HCHO/BC ratio in Europe? Please explain.

We agree that a more detailed discussion was missing from the previous version. We now provide possible explanations for the observed relationships and their differences between regions and in city plumes. In the revised version, we discuss the factors involved in the variability of gas-aerosol relationships, for example for the CO-BC relationship in section 3.1 (L 153-156 in the new version):

*"The CO-BC relationship depends first on the CO/BC ratio at emission in anthropogenic and fire emissions, second on the dilution and mixing of air masses, and on third the deposition. The CO average chemical lifetime is of one to three months, thus there is a different regional background*

*concentrations in Europe and Asia, while this is not the case for BC with a lifetime of a few weeks (e.g., Seinfeld and Pandis, 1997)."*

The results of the model simulations have been removed from the revised version because a dedicated article would be needed to understand the response of the models for the HCHO/BC ratio in both regions.

16、 Lines 236-237: Please be careful in discussion. It is not proving that the model bias primarily comes from emission inventory.

In this case, the sentence about models has been removed. However, this sentence was not claiming, but rather suggesting that the model bias comes primarily from the emission inventory, this sentence from the previous version is: *"We note the high R-value of BC with SO2 for the model ensemble, which suggests an overestimation of this ratio in the emission inventory. "*

This interpretation of model overestimation is related to the fact that if gas-aerosol ratios are not correct in the model, they cannot be correct in the atmosphere. On the contrary, if gas-aerosol ratios are accurate at emissions in the model, it may be inaccurate in the atmosphere because it is modified by chemistry, deposition, and mixing with different types of pollution.

In general, we are careful to formulate hypotheses that can be supported by our available data and analyses, and to discuss different possible explanations for the observed relationships.

17、 Lines 246-249 and Table 2: Why do the R-values of BC with the four trace gases (CO, HCHO, $NO_2$, and $SO_2$) decrease in city plumes? Additionally, why does the R-value of BC with $O_3$ increase in city plumes? Is this related to the aging of carbonaceous aerosols during transport? Please explain.

In the revised version, city plumes are investigated separately for the European and Asian campaigns (as already mentioned in specific comment #10). The O3-BC relationship is interpreted in Section 3.1 (L 166-167 in the new version):

*"We note that O3 is more strongly correlated with BC in city plumes, suggesting an indirect link through the emission of O3 precursors associated with BC emission."*

18、 Lines 262-267 and Table 3: Why are the R-values of the observed OA with the trace gases (CO, HCHO, $NO_2$, and $SO_2$) higher in Europe than those in Asia? Additionally, why is the R-value of the observed OA with $O_3$ lower in Europe than that in Asia? It requires more comparisons with previous studies and discussions. I recommend merging the discussion with Specific Comment #10.

Regarding the O3-OA relationship, our results show that this relationship stands out in city plumes (L 197-200 in the new version):

*"The strong O3-OA relationship in city plumes may be related to the co-emission of BC, OA and CO along with O3 precursors and SOA formation (e.g., Turpin and Huntzicker, 1991; Jimenez et al., 2009; McMeeking et al., 2011; Zhang et al., 2015; Yoon et al., 2021)."*

Moreover, we have added a detailed discussion of the differences between the two campaigns, as already mentioned in specific comment #10.

19、 Lines 269-271: Why is the observed CO/OA ratio in Europe much lower than in Asia? Additionally, why is the observed HCHO/OA ratio similar in Europe and Asia? It requires more comparisons with previous studies and discussions. I recommend merging the discussion with Specific Comment #12.

Regarding the HCHO-OA relationship, our results show that this relationship our results show that this relationship is strong in Europe but weak in Asia in city plumes (L 184-187 in the new version):

*"Liao et al. (2019) have shown that the HCHO-OA relationship is strong, reaching R²-values values up to 0.35, and that the HCHO/OA ratio has different values depending on the environment. We note that the HCHO-OA relationship is strong in Europe, so that HCHO is as relevant a proxy for OA as CO, whereas this is not the case in Asia (R² = 0.44 in Europe compared to 0.07 in Asia)."*

20、 Lines 284-286: Could you provide any comments on the strong linear relationship of CO/OA in regional simulations compared to observations?

The results from model simulations have been removed from the revised version.

In general, the relevant linear relationships identified with observations are stronger in the models. In addition to strong linear CO-OA relationships, strong linear CO-BC relationships are modeled in regional simulations. We interpreted the strong modeled CO-OA relationships as being mainly related to emissions and not to atmospheric processes, as already mentioned in specific comment #16. However, analysis of the results of model simulations would require further analysis in a specific article.

21、 Lines 295-297 and Table 4: Why do the R-values of OA with the four trace gases (CO, HCHO, $NO_2$, and $SO_2$) decrease in city plumes? Why does the R-value of OA with $O_3$ increase in city plumes? I recommend merging the discussion with Specific Comment #17.

The previous analysis of the gas-aerosol relationship in city plumes did not separate the European and Asian campaigns. The new analysis including this separation shows that, with the exception of CO-OA and O3-OA, the other linear gas-OA relationships are not relevant, as their R² values are less than 0.3 (new section 3.2).

22、 Section 6 : The discussions are critical, but the majority of this section lacks support from current analysis (see Major Comment #2).

We agree that discussions on gas-aerosol emission ratios are critical and could be addressed by an entire article dedicated to improving inventories on the basis of an assessment of these ratios using model simulations along with aircraft and satellite observations.

In the revised version, we have removed the analysis of emission pollutant ratios, as they are not directly comparable with atmospheric concentration pollutant ratios. Differences in these ratios should be related to lifetime (due to transport, deposition and chemical processes) which would modify and disconnect emission ratios from observed ratios in the atmosphere, as already mentioned in specific comment #16.

23、 Lines 377-389: The analysis of city plumes is too brief. Additionally, there is little analysis of satellite-based estimation of aerosols, but extensive results of model evaluations (see Major Comment #3). I recommend integrating the model evaluations into the scientific questions.

As we have focused this revised version on the important results which concern the observations, models are not integrated into the scientific question explored in this revised version. Rather we propose new questions, which are clearly answered with our results only based on observations, as already mentioned in our response to specific comment major comment #3.

Our approach is to examine the statistical relationships and, if possible, discuss whether they are consistent with the scientific literature. We then move beyond linear relationships with a machine learning approach. This evaluation of the statistical relationships between carbonaceous aerosol and trace gas concentrations is an important step towards our overarching goal, which is to estimate carbonaceous aerosol using satellite observations of trace gases. This is explained in the last paragraph of the conclusion (L 346-352 in the new version):

*"This study suggests that satellite retrievals of trace gases may be relevant for estimating carbonaceous aerosols, especially since the performance of regional air quality models for simulating carbonaceous aerosols is generally weak, reaching R²-values of ≈ 0.7 (Mircea et al., 2019). Machine learning approaches trained with ground-based measurements of BC and OA concentrations as predicted variables, together with satellite retrievals of trace gases as predictors, could provide BC and OA concentrations with high accuracy. In addition, ground-based PM2.5 concentration and satellite retrievals of Aerosol Optical Depth could be included as predictors, which could increase the accuracy of aerosol composition estimation, especially focusing on megacities where pollutant emissions are strong, continuous, and localized."*

**Bibliography (not in the manuscript)**

Menut, L., Mailler, S., Siour, G., Bessagnet, B., Turquety, S., Rea, G., Briant, R., Mallet, M., Sciare, J., Formenti, P., & Meleux, F. (2015). Ozone and aerosol tropospheric concentrations

variability analyzed using the ADRIMED measurements and the WRF and CHIMERE models. *Atmospheric Chemistry and Physics*, *15*(11), 6159–6182. https://doi.org/10.5194/acp-15-6159-2015

Petetin, H., Sciare, J., Bressi, M., Gros, V., Rosso, A., Sanchez, O., Sarda-Estève, R., Petit, J.-E., & Beekmann, M. (2016). Assessing the ammonium nitrate formation regime in the Paris megacity and its representation in the CHIMERE model. *Atmospheric Chemistry and Physics*, *16*(16), 10419–10440. https://doi.org/10.5194/acp-16-10419-2016

Deroubaix, A., Flamant, C., Menut, L., Siour, G., Mailler, S., Turquety, S., Briant, R., Khvorostyanov, D., & Crumeyrolle, S. (2018). Interactions of atmospheric gases and aerosols with the monsoon dynamics over the Sudano-Guinean region during AMMA. *Atmospheric Chemistry and Physics*, *18*(1), 445–465. https://doi.org/10.5194/acp-18-445-2018

Deroubaix, A., Menut, L., Flamant, C., Brito, J., Denjean, C., Dreiling, V., Fink, A., Jambert, C., Kalthoff, N., Knippertz, P., Ladkin, R., Mailler, S., Maranan, M., Pacifico, F., Piguet, B., Siour, G., & Turquety, S. (2019). Diurnal cycle of coastal anthropogenic pollutant transport over southern West Africa during the DACCIWA campaign. *Atmospheric Chemistry and Physics*, *19*(1), 473–497. https://doi.org/10.5194/acp-19-473-2019

Flamant, C., Deroubaix, A., Chazette, P., Brito, J., Gaetani, M., Knippertz, P., Fink, A. H., de Coetlogon, G., Menut, L., Colomb, A., Denjean, C., Meynadier, R., Rosenberg, P., Dupuy, R., Dominutti, P., Duplissy, J., Bourrianne, T., Schwarzenboeck, A., Ramonet, M., & Totems, J. (2018). Aerosol distribution in the northern Gulf of Guinea: local anthropogenic sources, long-range transport, and the role of coastal shallow circulations. *Atmospheric Chemistry and Physics*, *18*(16), 12363–12389. https://doi.org/10.5194/acp-18-12363-2018